# Two Sides of Meta-Learning Evaluation:
# In vs. Out of Distribution

**Amrith Setlur**[1][*]     **Oscar Li**[2*]     **Virginia Smith**[2]

asetlur@cs.cmu.edu   oscarli@cmu.edu   smithv@cmu.edu

[1]Language Technologies Institute     [2]Machine Learning Department

School of Computer Science, Carnegie Mellon University

## Abstract

We categorize meta-learning evaluation into two settings: *in-distribution* [ID], in which the train and test tasks are sampled *iid* from the same underlying task distribution, and *out-of-distribution* [OOD], in which they are not. While most meta-learning theory and some FSL applications follow the ID setting, we identify that most existing few-shot classification benchmarks instead reflect OOD evaluation, as they use disjoint sets of train (base) and test (novel) classes for task generation. This discrepancy is problematic because—as we show on numerous benchmarks—meta-learning methods that perform better on existing OOD datasets may perform significantly worse in the ID setting. In addition, in the OOD setting, even though current FSL benchmarks seem befitting, our study highlights concerns in 1) reliably performing model selection for a given meta-learning method, and 2) consistently comparing the performance of different methods. To address these concerns, we provide suggestions on how to construct FSL benchmarks to allow for ID evaluation as well as more reliable OOD evaluation. Our work[†] aims to inform the meta-learning community about the importance and distinction of ID vs. OOD evaluation, as well as the subtleties of OOD evaluation with current benchmarks.

## 1  Introduction

Meta-learning considers learning algorithms that can perform well over a distribution of tasks [19, 37]. To do so, a meta-learning method first learns from a set of tasks sampled from a training task distribution (*meta-training*), and then evaluates the quality of the learned algorithm using tasks from a test task distribution (*meta-testing*). The test task distribution can be the same as the training task distribution (a scenario we term *in-distribution* generalization evaluation or ID evaluation) or a different task distribution (*out-of-distribution* generalization evaluation or OOD evaluation).

In this work, we argue that there is a need to carefully consider current meta-learning practices in light of this ID vs. OOD categorization. In particular, meta-learning is commonly evaluated on few-shot learning (FSL) benchmarks, which aim to evaluate meta-learning methods' ability to learn sample-efficient algorithms. Current benchmarks primarily focus on image classification and provide training tasks constructed from a set of train (base) classes that are completely disjoint and sometimes extremely different from the test (novel) classes used for test tasks. As we discuss in Section 3, this design choice imposes a natural shift in the train and test task distribution that makes current benchmarks reflective of OOD generalization. However, there are a number of reasons to also consider the distinct setting of ID evaluation. First, whether in terms of methodology or theory, many works motivate and analyze meta-learning under the assumption that train and test tasks are sampled *iid* from the same distribution (see Section 2). Second, we identify a growing number of applications, such as federated learning, where there is in fact a need for sample-efficient algorithms

---

[*]Authors contributed equally to this paper.

[†]Code available at https://github.com/ars22/meta-learning-eval-id-vs-ood.

35th Conference on Neural Information Processing Systems (NeurIPS 2021).

that can perform ID generalization. Crucially, we show across numerous benchmarks that methods that perform well OOD may perform significantly worse in ID settings. Our results highlight that it is critical to clearly define which setting a researcher is targeting when developing new meta-learning methods, and we provide tools for modifying existing benchmarks to reflect both scenarios.

Beyond this, we also re-examine current OOD FSL benchmarks and analyze how the shift in the train and test task distributions may impact the reliability of OOD evaluations. We point out two concerns which we believe are not widely considered in the meta-learning community. First, unlike areas such as domain generalization where model selection challenges are more widely discussed [18, 23], we conduct to the best of our knowledge the first rigorous study demonstrating the difficulty of model selection due to the shift in the validation and test task distributions in FSL benchmarks. Second, because the OOD scenario in meta-learning does not assume a specific test task distribution, there is room for different test distributions to be used for evaluation. We show that comparing which meta-learning method performs better can be unreliable not only over different OOD FSL benchmarks, but also within a single benchmark depending on the number of novel classes.

Our main contributions are: **i)** We clearly outline both ID and OOD FSL evaluation scenarios and explain why most popular FSL benchmarks target OOD evaluation (Section 3). **ii)** We provide realistic examples of the ID scenario and show that the performance of popular meta-learning methods can drastically differ in ID vs. OOD scenarios (Section 4). **iii)** For existing OOD FSL benchmarks, we highlight concerns with a) current model selection strategies for meta-learning methods, and b) the reliability of meta-learning method comparisons (Section 5). **iv)** To remedy these concerns, we suggest suitable modifications to the current FSL benchmarks to allow for ID evaluation, and explain how to construct FSL benchmarks to provide more reliable OOD evaluation. Our hope in highlighting these evaluation concerns is for future researchers to consider them when evaluating newly proposed meta-learning methods or designing new FSL benchmarks.

## 2   Related Work

**Current FSL benchmarks.** A plethora of few-shot image classification benchmarks (*e.g., mini*-ImageNet (*mini* in short) [42], CIFAR-FS [4]) have been developed for FSL evaluation. These benchmarks typically provide three disjoint sets of classes (base, validation, novel) taken from standard classification datasets, *e.g.,* ImageNet or CIFAR-100 [36, 32, 42]. Training, val, and test tasks are then constructed from these classes respectively, which, as we discuss in Section 3, *can induce a shift in their corresponding task distributions*. Distribution mismatch can be particularly large with non-random splits created at the super class level, *e.g.,* FC-100 [30], or dataset level, *e.g.,* Meta-Dataset [40]. Recently, Arnold and Sha [2] propose an automated approach to construct different class splits from the same dataset to allow for varying degrees of task distribution shifts; Triantafillou et al. [41] separate the distribution shifts on Meta-Dataset into weak vs. strong generalization depending on whether the novel classes are taken from the used training datasets or not. Both works find that the meta-learning methods that perform better in one distribution shift scenario might perform worse in another, providing further evidence to our OOD performance comparison inconsistency argument in Section 5.2. Beyond these canonical ways of task construction through a set of classes, Ren et al. [35] propose new benchmarks in a new flexible few-shot learning (FFSL) setting, where the aim is to classify examples into a context instead of an object class. During testing, they perform OOD evaluation on unseen contexts. Inspired by their approach, we also provide an FSL benchmark where tasks are specified by contexts (Section 4), though we differ by exploring ID evaluation.

**Mismatch between meta-learning theory/methodology and evaluation.** Despite theoretical works which analyze meta-learning OOD generalization [11, 13], there are many theoretical meta-learning works [e.g., 1, 3, 8, 22] that first assume the train and test tasks are *iid* sampled from the same distribution despite validating their analyses on OOD FSL benchmarks. Additionally, several popular meta-learning methods [25, 16, 31] that are motivated in the ID scenario are largely evaluated on OOD benchmarks (see Appendix A). Prior work of Lee et al. [24] explores ID vs. OOD; however they treat the FSL setup as ID when the disjoint base and novel classes are from the same dataset and OOD only when they are not (*e.g.,* base from ImageNet, novel from CUB). We emphasize that even the disjointedness of base and novel from the same dataset can create a task distribution shift and hence unlike [24] we do not consider current FSL benchmarks (like *mini*) to be ID (Section 3).

**FSL ID evaluation.** We are unaware of any FSL classification benchmarks that are explicitly advertised for ID evaluation. As we discuss in Section 4, a growing number of works [7, 12, 20, 22, 27]

use meta-learning for personalized federated learning, but do not clearly discuss the in-distribution nature of these benchmarks nor how they differ from standard FSL benchmarks. In recent work, Chen et al. [10] extend current FSL benchmarks to evaluate their proposed method both ID and OOD, but only to further improve OOD performance on the original FSL benchmark's novel classes. Our work uses a similar setup for constructing ID evaluations with current OOD FSL benchmarks, but focuses on showing that certain meta-learning methods/design choices can improve OOD performance at the cost of ID performance. Prior work [17, 34] on incremental few-shot/low-shot learning also explores performance on both base and novel classes simultaneously. However, they differ in their methodology, as they use supervised learning (not meta-learning) to classify over the base classes.

**OOD evaluation in other fields.** Train and test distribution shifts are also found in domain generalization [5, 29] where the goal is to find a model that works well for a different test environment. Due to this shift, Gulrajani and Lopez-Paz [18] specifically discuss difficulties in performing model selection in domain generalization benchmarks. They argue that it is the responsibility of the method designer (not benchmark designer) to determine model selection strategies for their method, and propose several model selection methods, mainly targeting hyperparameter selection. Unlike domain generalization, FSL benchmarks often have a pre-determined disjoint set of validation classes to construct validation tasks, so the need for developing model selection methods may be less obvious. In Section 5, motivated by [18], we explore strategies for model selection for meta-learning. However, in contrast to [18], we focus on algorithm snapshot selection for meta-learning, which is required by hyperparameter selection as a subroutine (exact definitions see Section 5).

## 3 FSL benchmarks: Background & Focus on OOD evaluation

**Background and notation.** In this work, we employ a general definition of a meta-learning FSL task: a task $\mathcal{T}$ is a distribution over the space of support and query dataset pairs $(S, Q)$, where $S, Q$ are two sets of examples from an example space $\mathcal{X} \times \mathcal{Y}$. The support set $S$ is used by an algorithm to produce an adapted model which is then evaluated by the corresponding query set $Q$. Each time we interact with a task $\mathcal{T}$, an $(S, Q)$ pair is sampled *iid* from $\mathcal{T}$, the mechanism for which depends on the specific application; we provide multiple examples below. As discussed in Section 1, meta-learning aims to learn an algorithm over a distribution of tasks $\mathbb{P}(\mathcal{T})$. During meta-training, we assume access to $N$ pairs of $\{(S_i, Q_i)\}_{i \in [N]}$ sampled from a *training task distribution*[‡] $\mathbb{P}_{\mathrm{tr}}(\mathcal{T})$ in the following way: first, $N$ tasks are sampled *iid* from the training task $\mathcal{T}_i \sim \mathbb{P}_{\mathrm{tr}}(\mathcal{T})$; then, for each task $\mathcal{T}_i$, a support query pair $(S_i, Q_i) \sim \mathcal{T}_i$ is sampled *iid*. During meta-testing, we assume there is a *test task distribution* $\mathbb{P}_{\mathrm{te}}(\mathcal{T})$ where fresh $(S, Q)$ samples are similarly sampled based on $P_{\mathrm{te}}(\mathcal{T})$. We define a learning scenario to be in-distribution (ID) if $\mathbb{P}_{\mathrm{tr}} = \mathbb{P}_{\mathrm{te}}$ and out-of-distribution (OOD) if $\mathbb{P}_{\mathrm{tr}} \neq \mathbb{P}_{\mathrm{te}}$. Whenever $\mathbb{P}_{\mathrm{tr}} = \mathbb{P}_{\mathrm{te}}$, the induced train and test marginal distributions of $(S, Q)$ are also identical.

**Construction of $(S, Q)$ pairs in FSL.** Most popular FSL benchmarks share a similar structure: they provide three disjoint sets of classes: base classes $\mathcal{C}_B$, validation classes $\mathcal{C}_V$, and novel classes $\mathcal{C}_N$, where any class $c$ in these sets specifies a distribution $\mathbb{P}_c$ over the example space $\mathcal{X}$. An $n$-way $k$-shot $q$-query task $\mathcal{T}_c$ in these benchmarks is specified by a length $n$ non-repeated class tuple $\boldsymbol{c} = (c_1, \ldots, c_n)$ where $\boldsymbol{c} \in [\mathcal{C}^n] := \{(d_1, \ldots, d_n) \in \mathcal{C}^n : d_i \neq d_j, \forall i \neq j\}$. $\mathcal{T}_c$ generates random $(S, Q)$ pairs in the following way: For every class $c_i$, $k$ support examples $S_i \sim (\mathbb{P}_{c_i})^k$ and $q$ query examples $Q_i \sim (\mathbb{P}_{c_i})^q$ are sampled. The support and query set is formed by the union of such labelled examples from each class: $S = \cup_{i=1}^n \{(x, i), \forall x \in S_i\}$, $Q = \cup_{i=1}^n \{(x, i), \forall x \in Q_i\}$. By specifying the base and novel classes, the FSL benchmark has provided a collection of tasks for training $\{\mathcal{T}_c : \boldsymbol{c} \in [\mathcal{C}_B^n]\}$ and test $\{\mathcal{T}_c : \boldsymbol{c} \in [\mathcal{C}_N^n]\}$. These sets can be extremely large. For example, in *mini*, which has 64 base classes, the total number of 5-way training tasks is $\frac{64!}{(64-5)!} \approx 9.1 \times 10^8$. However, it is not explicitly specified what underlying task distribution $\mathbb{P}(\mathcal{T})$ generates these sets of tasks. We believe this may have led prior work to discuss FSL benchmarks in the context of ID evaluation [e.g., 1, 3, 8], contrary to what we outline below.

**Current FSL benchmarks target OOD evaluation.** We now informally discuss our reasoning for arguing that current FSL benchmarks reflect OOD evaluation (we provide a more formal proof by contradiction in Appendix B). In particular, if the training and test tasks in FSL benchmarks are indeed *iid* sampled from the same underlying task distribution, then this underlying distribution must be induced by a distribution over class tuples of an even larger class set $\mathcal{C}_L$ ($\mathcal{C}_L \supseteq (\mathcal{C}_B \cup \mathcal{C}_V \cup \mathcal{C}_N)$).

---

[‡]We note that "training task distribution" here refers to the true underlying distribution of training tasks, not the empirical distribution supported over the finite set of sampled training tasks.

We consider the following dichotomy:

i) when $|\mathcal{C}_L| = \mathcal{O}(nN)$: In this case, the total number of classes $nN$ covered by the sampled tasks (counting repetition) is greater than the number of underlying classes. Then with high probability, both the training and test tasks would each cover a significant portion of all the classes in $\mathcal{C}_L$, making it extremely unlikely to have an empty intersection as in the current FSL benchmarks.

ii) when $|\mathcal{C}_L| = \Omega(nN)$: In this alternative case, the total number of classes (even counting repetitions) used by sampled tasks is still smaller than the number of underlying classes $|\mathcal{C}_L|$. Thus the sampled tasks cannot cover all the underlying classes. Under this regime, the number of classes covered by the training tasks alone would scale linearly with $N$, as repeating an already-seen class in a new task sample is relatively rare. Since FSL benchmarks typically use a large number of training tasks during meta-training ($N > 10^3$), it is improbable that all the training tasks would together only cover a very low number of classes (64 in the case of *mini*).

**Randomized class partitions do not imply randomized task partitions.** Another issue that may cause some to view the current FSL benchmarks as performing ID evaluation is that in some of these benchmarks, the base, val, novel classes are random partitions of *iid* drawn classes from a class level distribution (specifically *mini*ImageNet, CIFAR-FS; but not FC-100, *tiered*ImageNet as the classes are not partitioned randomly). The logic here is that in standard machine learning practice, randomly partitioning *iid* sampled data points into train and test guarantees that the train and test samples are drawn *iid* from the same underlying distribution. However, it is important to notice that *the first class citizen in common FSL benchmarks is not a class, but a task* (represented by a class tuple). So, only a randomized partition of *iid* sampled class tuples would guarantee in-distribution sampling.

**How can we view $\mathbb{P}_{\mathrm{tr}}, \mathbb{P}_{\mathrm{te}}$ in common FSL benchmarks?** Based on the discussion above, we need to view train and test tasks in current FSL benchmarks as coming from different distributions, *i.e.,* $\mathbb{P}_{\mathrm{tr}} \neq \mathbb{P}_{\mathrm{te}}$. In order to ensure that both sets of tasks are still sampled *iid* from their respective distributions, it is convenient to view the train/test tasks as being *iid* sampled from a uniform distribution over all possible train/test class tuples induced by $\mathcal{C}_B/\mathcal{C}_N$ *i.e.,* $\mathbb{P}_{\mathrm{tr}} = \mathbb{P}_{\mathcal{C}_B} := \mathrm{Unif}(\{\mathcal{T}_{\boldsymbol{c}} : \boldsymbol{c} \in [\mathcal{C}_B^n]\})$ and test $\mathbb{P}_{\mathrm{te}} = \mathbb{P}_{\mathcal{C}_N} := \mathrm{Unif}(\{\mathcal{T}_{\boldsymbol{c}} : \boldsymbol{c} \in [\mathcal{C}_N^n]\})$ — a view which we will adopt in the rest of the paper.

## 4   Evaluating In-Distribution Performance

Although (as discussed in Section 3) current FSL benchmarks target OOD evaluation, we now explore example applications where ID generalization is instead required, and provide easily constructible benchmarks mirroring these scenarios. As we will show, this distinction is important because meta-learning methods may perform markedly different in ID vs OOD scenarios.

***Example 1* (Federated Learning):** Multiple works [7, 12, 20, 22, 27] have considered applying meta-learning methods in federated learning, in which the goal is to learn across a distributed network of devices [28, 26]. Meta-learning can produce personalized models for unseen devices/users, improving over a single globally-learned model's performance. In this setting, a popular benchmark is the FEMNIST [6] handwriting recognition dataset. For FEMNIST, we assume there exists a distribution of writers $\mathbb{P}(\texttt{id})$ in a federated network where each writer (with a unique $\texttt{id}$) is associated with a few-shot classification problem to recognize over the different character classes the writer has written for. We associate each writer $\texttt{id}$ with a task $\mathcal{T}_{\texttt{id}}$ which randomly generates a support set with one random example per class and a query set with varying number of random examples per class.

**ID evaluation on FEMNIST.** We are given a total of $\sim 3500$ writers sampled *iid* from $\mathbb{P}(\texttt{id})$ and we randomly partition them into a $2509/538/538$ split for training, validation, and test tasks, following similar practices used in prior FL work [20, 7]. Note that this random split is performed at the task/$\texttt{id}$ level. As such, we can treat the training and test tasks as being sampled *iid* from the same task distribution, unlike current FSL benchmarks.

***Example 2* (Online Recommendation):** Ren et al. [35] propose the use of *Zappos* [43] dataset as a meta-learning benchmark where each task is a binary classification of shoe images into an attribute context. This mimics an online shopping recommendation problem, where each user has different shoe preferences based on specific shoe attributes (hence a single global predictive model would not perform well), and the recommendation system must quickly learn a user's likes/dislikes through a few interactions. In this simplified setup, we fix a universal set of shoe attributes $\mathcal{A}$, and each user's preference is represented by a specific pair of unique attributes $\boldsymbol{a} = (a_1, a_2)$, $a_1 \neq a_2$. A task $\mathcal{T}_{\boldsymbol{a}}$ representing a user with attribute preference $\boldsymbol{a}$ generates 2-way $k$-shot $q$-query $(S, Q)$ pair by *iid* sampling $k + q$ examples both from the set of images that carry both attributes in $\boldsymbol{a}$ (positive

Table 1: Ranking in () of meta-algorithms' test performance on **i)** ID benchmarks FEMNIST, Zappos-ID (with either 1000 or 50 training tasks); and **ii)** OOD FSL benchmark *mini*ImageNet.

| Dataset / | FEMNIST | Zappos-ID | | | *mini*ImageNet |
|---|---|---|---|---|---|
| Method | *w1s | 2w10s | 2w5s (1000 train tasks) | 2w5s (50 train tasks) | 5w5s |
| PN | [1]$94.72 \pm 0.41\%$ | [1]$88.40 \pm 0.13\%$ | [1]$86.58 \pm 0.15\%$ | [1]$77.67 \pm 0.17\%$ | [3]$76.22 \pm 0.14\%$ |
| Ridge | [1]$94.71 \pm 0.42\%$ | [2]$88.01 \pm 0.14\%$ | [2]$85.56 \pm 0.16\%$ | [2]$74.75 \pm 0.16\%$ | [2]$77.20 \pm 0.15\%$ |
| SVM | [3]$94.22 \pm 0.45\%$ | [3]$87.75 \pm 0.14\%$ | [3]$85.12 \pm 0.16\%$ | [3]$74.06 \pm 0.17\%$ | [1]$77.72 \pm 0.15\%$ |
| FO-MAML | N/A | [4]$81.90 \pm 0.14\%$ | [4]$80.14 \pm 0.15\%$ | [4]$69.85 \pm 0.18\%$ | [4]$75.96 \pm 0.17\%$ |

examples) and from the set that does not (negative examples). Our task distribution is a uniform distribution over tasks of all attribute pairs $\mathbb{P}_{\mathcal{A}}(\mathcal{T}) = \text{Unif}(\{\mathcal{T}_{\boldsymbol{a}} : \boldsymbol{a} \in [\mathcal{A}^2])\})$.

**ID evaluation on Zappos.** Unlike in [35], where Zappos is used to measure OOD performance by having disjoint train and test attributes $\mathcal{A}_{\text{tr}} \cap \mathcal{A}_{\text{te}} = \phi$, in this work we use Zappos for ID evaluations by *iid* sampling both meta-train and meta-test tasks from the same distribution $\mathbb{P}_{\mathcal{A}}(\mathcal{T})$. Through this modification of Ren et al.'s setup, we sample 1000 / 50 attribute pairs from an attribute set $|\mathcal{A}| = 36$ to construct 1000 / 50 training tasks (each with a randomly sampled $(S, Q)$) and evaluate ID performance on another 25000 test tasks sampled in the same way. Our evaluation setup captures a realistic setting where the goal is to learn an algorithm that can generalize to the entire online shopper population despite being trained only on a randomly chosen subset of shoppers.

**Remark.** Even with ID evaluation it is possible to encounter unseen classes/attributes in meta-test tasks, specifically when the number of meta-train tasks is smaller than the number of underlying classes/attributes (Section 3). However, a suitable number of *iid* sampled meta-train tasks is needed to ensure good performance, which would naturally encompass a larger set of meta-train classes than those considered by OOD FSL benchmarks. For example, there are still 16 attribute pairs from the test tasks that are unseen in the 1000 training tasks on Zappos-ID, but the dataset is still in-distribution since the sampling distributions of train and test attribute pairs (and thus of tasks) are identical.

**ID benchmark results.** We evaluate the ID performance of four popular meta-learning methods: Prototypical Networks (PN) [39], MetaOptNet-SVM (SVM) [25], MetaOptNet-Ridge Regression (RR) [25, 4] and FOMAML [15] on our identified ID FSL benchmarks (Table 1). Since FEMNIST's tasks have varying number of ways, FOMAML cannot be directly used due to the logit layer shape mismatch. We note that the performance order of the four methods are consistent on all three ID benchmarks yet surprisingly **almost completely opposite** to the performance order observed on the OOD benchmark *mini* (except for FOMAML). In terms of the actual performance differences, we notice that the ID performance advantage of PN over SVM becomes particularly large ($> 3\%$) when we reduce of the number of training tasks for Zappos-ID to 50; in contrast, on the OOD benchmark *mini*, SVM instead outperforms PN by 1.5% (a significant number as many newly proposed meta-learning methods often only report improvements over previous methods by $1.5\%$ or less). These performance differences make it clear that the performance ranking flips between ID and OOD indeed exist, and as a result, these common OOD FSL benchmarks (like *mini*) cannot be used to compare ID performance without modifications, giving further evidence to the danger of such practices (see Section 2). To further understand this phenomenon, we propose a way to also enable ID performance evaluation over these common OOD FSL benchmarks and see if there still exists a difference in ID, OOD performance orders.

**Modifying OOD FSL benchmarks for ID evaluation.** From our previous discussion, we have shown that we can think of FSL training tasks as being sampled *iid* from the task distribution $\mathbb{P}_{\mathcal{C}_B} \coloneqq \text{Unif}(\{\mathcal{T}_{\boldsymbol{c}} : \boldsymbol{c} \in [\mathcal{C}_B^n]\})$. To conduct an in-distribution evaluation, we need to sample fresh test tasks *iid* from $\mathbb{P}_{\mathcal{C}_B}$. For a freshly sampled $\mathcal{T}_{\boldsymbol{c}} \sim \mathbb{P}_{\mathcal{C}_B}$, we also need to *iid* sample a fresh support query pair $(S, Q) \sim \mathcal{T}_{\boldsymbol{c}}$. To ensure independent sampling from the already seen meta-training $(S, Q)$ pairs, we need to introduce new examples from $\mathbb{P}_c$ for each class $c \in \mathcal{C}_B$. Thus we construct slightly modified versions of four common FSL benchmarks **i)** *mini*ImageNet-Mod (*mini*-M) [42], in which we find ($\approx 700$) unused examples (from ImageNet) for each base class and use them to evaluate the performance over $\mathbb{P}_{\mathcal{C}_B}$. Here the original 600 examples of each base class are still only used for meta-training. **ii)** CIFAR-FS-Mod (*cifar*-M) [30], FC-100-Mod (FC-M) [4], and *tiered*ImageNet-Mod (*tiered*-M) [33]: As we don't have additional samples for base classes, we randomly partition each base class's current examples into an approximate $80/20$ split where the training tasks are constructed using the former and the latter is reserved for ID evaluation.

**ID vs. OOD conflicts still exist.** In addition to evaluating the four aforementioned meta-learning methods, we also consider two supervised pretraining methods: the supervised learning baseline (SB)

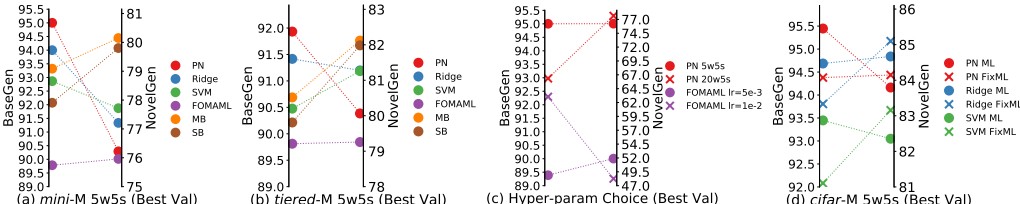

Figure 1: We show the BaseGen and NovelGen performance tradeoff (for best validation snapshots): over the choice of a set of four meta-learning and two supervised pre-training methods on *mini*-M (a) and *tiered*-M (b); over the number of ways to train PN on *mini*-M and different learning rates to train FOMAML on FC-M (c); over the use of FIX-ML $(S, Q)$ generation strategy or not (ML) with SVM, RR and PN on *cifar*-M in (d).

[9] and Meta-Baseline (MB) [10]. Both of these methods have been shown to outperform many meta-learning methods on the current OOD FSL benchmarks. For succintness, we call the generalization performance over the training task distribution $\mathbb{P}_{\mathcal{C}_B}$ *BaseGen* and performance over $\mathbb{P}_{\mathcal{C}_N}$ *NovelGen* (we will refer to performance on validation tasks from $\mathbb{P}_{\mathcal{C}_V}$ *ValGen* in a later section). We plot the BaseGen and NovelGen performances of the aforementioned methods on *mini*ImageNet-Mod and *tiered*ImageNet-Mod in Figure 1(a)(b) (other datasets see Appendix E Figure 4(a)(b)), with a dotted line connecting BaseGen and NovelGen value of same learned algorithm snapshot of a meta-learning method. We see that the **ID/OOD performance order flips still exist within current FSL benchmarks themselves** (the dotted lines of different methods cross each other frequently), showing that **the issue of improving OOD at the expense of ID is a common realistic phenomenon for multiple benchmarks**. In addition, despite outperforming all meta-learning methods on NovelGen, the non-meta-learning methods SB and MB cannot beat the best meta-learning methods on BaseGen, which demonstrates their restrictive advantage only in the OOD setting. More broadly speaking, these OOD FSL datasets are constructed with a belief that there exists a commonality between the training and test task distributions so that an algorithm capturing this commonality using the training distribution alone would also generalize to the test tasks. Thus the phenomenon of improving OOD while sacrificing ID means the algorithm has in some sense failed to learn the essence/commonality from the given training tasks. Additionally, we see that the *BaseGen ranking of the four meta-learning methods over these common FSL bechmarks are exactly the same as the ranking over our two newly proposed ID benchmarks in Table 1*. We suggest that researchers who want to also test on their proposed methods' ID generalization performance can perform BaseGen evaluation method in our proposed way as it is a simple addition to their existing NovelGen evaluation setup.

**Certain OOD training choices might harm ID generalization.** In addition to checking the discrepancy of ID vs OOD generalization comparison of different meta-learning methods, we now ask, for a given meta-learning method, whether the meta-training choices found to be most effective on NovelGen would still be optimal for BaseGen.

i) **Meta-training hyperparameters:** In Figure 1(c) we see that a higher learning rate when training FOMAML finds algorithms that have higher BaseGen, whereas lower learning rates are better for NovelGen. Additionally, we see that the proposed technique of training with more ways for PN in [39] can lead to better NovelGen performance but worse BaseGen performance than the 5-way trained PN whose training and test task configurations match.

ii) **Meta-training $(S, Q)$ generation alternatives:** It was found in [38] that always using the same support examples for every base class when constructing $S$ (FIX-ML) can improve NovelGen performance for several meta-learning methods over multiple OOD FSL benchmarks. However, restricting the training $(S, Q)$'s diversity sampled from $\mathbb{P}_{\mathcal{C}_B}$ seems counter-intuitive and we wouldn't expect to improve the in-distribution generalization BaseGen. In Figure 1(d), we indeed see that FIX-ML only improves NovelGen performance at the expense of BaseGen by training over a much less diverse set of tasks.

These two observations above caution us that **some training techniques to boost test accuracy on FSL benchmarks might only work for the OOD scenario but not the ID scenario.**

## 5  Challenges With Out-of-Distribution Evaluation

After identifying some example benchmarks for ID evaluation, we come back to the current OOD FSL benchmarks to further examine some subtle challenges in OOD evaluation. Here, instead of focusing on the distinction between ID vs. OOD, we now look at some reliability and inconsistency problems

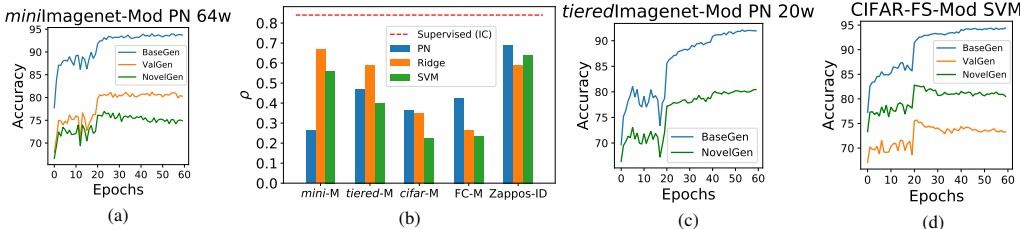

Figure 2: We plot the (Base, Val, Novel)Gen progression of 64(max)-way trained PN on *mini*-M in (a) and of SVM trained on *cifar*-M in (d). In (b) we compute the Kendall rank correlation coefficient ($\rho$) between the validation and test rankings of model snapshots for IC (trained on *cifar*) and algorithm snapshots for PN, SVM, RR on OOD datasets *mini*-M (last 40 epochs), *cifar*-M (last 40), FC-M (last 10), *tiered*-M (last 20) and ID dataset Zappos-ID (last 30); in (c) we show the BaseGen tracking NovelGen for 20-way trained PN on *tiered*-M.

*within the OOD evaluation itself*. In particular, we highlight two concerns: **1)** Despite providing a validation set of classes for task construction during meta-validation, it is not clear whether this is a reliable way to perform model selection. **2)** As there is no requirement on how similar the training and test task distributions need to be in the OOD setting, there can be inconsistencies in method comparisons when the evaluation setup is slightly modified.

## 5.1 Model Selection

To compare a set of meta-learning methods on the OOD test task distribution, one needs to *select* a representative algorithm learned by each method to be evaluated on the test task distribution. To do so, one should first decide what set of hyperparameter configurations to choose from for a given meta-learning method (we define the entire set of all hyperparameters for a training run as a *hyperparameter config*). For each such considered config, after its training is completed, we need to choose one of the algorithm snapshots saved during training to represent it (which we call *snapshot selection*). Then the set of hyperparameter configs are compared based on their respectively selected snapshots, and a single config is then chosen among them (which we term *hyperparameter selection*). This config's selected snapshot will represent the given meta-learning method to be finally evaluated on the test tasks. We refer to the combined problem of hyperparameter and snapshot selection as model selection. (See Appendix F.1 for a simplified example of snapshot and hyperparameter selection; see F.2 for the distinction between the commonly used technique *early-stopping* and snapshot selection.)

**Snapshot vs Hyperparameter selection.** If the snapshot selection strategy cannot reliably identify a snapshot with good test performance, it is possible that some hyperparameter configs will be unfairly represented by a mischosen snapshot, leading to erroneous hyperparameter selection. Thus we believe snapshot selection is a more fundamental problem and we focus our analyses on it (more on hyperparameter selection in Appendix F.5). In OOD FSL benchmarks, a reserved set of validation classes (disjoint from the set of novel classes) is provided, which as we have argued, provides tasks which are not sampled *iid* from the test task distribution. As a result, we ask: *is performing validation on the given validation tasks reliable, and, if not, are there other options?* In contrast, there is typically no need for such concerns in standard ID supervised learning, where the validation set is sampled *iid* from the test distribution.

### 5.1.1 *Option 1*: Snapshot selection tradition using ValGen.

By providing a set of validation classes, it has become the default practice for meta-learning methods to use ValGen performance for snapshot selection. However, because $\mathbb{P}_{\mathcal{C}_V}$ and $\mathbb{P}_{\mathcal{C}_N}$ are different tasks distributions, it is not clear whether a higher ValGen performance is *strongly correlated with* a higher NovelGen performance. In Figure 2(a), we plot the progression of ValGen and NovelGen of a 64-way trained PN on *mini*ImageNet-Mod 5w5s tasks. We notice that ValGen is consistently higher than NovelGen, indicating that meta-val performance is not an accurate estimator of NovelGen. More importantly, we see trendwise that while ValGen is generally non-decreasing, NovelGen starts to decrease after epoch 30. Thus the snapshot selected according to the best ValGen is not the snapshot with the best possible meta-test performance. In fact, this loss of NovelGen performance due to choosing the best ValGen model instead of the actual best NovelGen model can be particularly large, with values being 1.1% for SVM, 1.2% for RR, 0.7% for PN, and 0.9% for FOMAML on the FC-100 dataset. These performances losses for each method are especially concerning considering the differences among the best possible NovelGen performance of these different methods are often smaller than 1.5%.

**Ranking similarity analysis.** In light of the above observation, we ask a more general quantitative question: *How **similar** is the ranking of the training snapshots using meta-val performance (ValGen)* ***to** the ranking using the meta-test performance (NovelGen)?* To answer this, we compute the Kendall rank correlation coefficient[§] $\rho$ [21] between the ValGen and NovelGen rankings of algorithm snapshots trained on four OOD FSL benchmarks (Figure 2(b)) and our ID benchmark Zappos-ID whose validation and test tasks come from the same distribution $\mathbb{P}_{\mathcal{A}}$. More concretely, for each meta-learning method and dataset combination, we save the algorithm snapshots (one from each epoch) throughout meta-training and rank these algorithm snapshots according to their ValGen and NovelGen value respectively. Then $\rho$ is computed between these two rankings for this specific (meta-learning method,dataset) combination. For snapshot selection through ValGen to work reliably, we need $\rho$ to be close to 1. For context, we also compute $\rho$ for a standard supervised image classification problem (IC), where train, val and test examples are sampled from the same example-level distribution.

**Unreliabity of ValGen snapshot selection.** From Figure 2(b), we see that when using validation samples generated *iid* from the test distribution (Zappos-ID and supervised learning IC), the value of $\rho$ is consistently higher than the OOD benchmarks, indicating the validation performance can more reliably track the trend of test performance in the ID setting than on the OOD FSL benchmarks. In particular, for the *cifar*-M and FC-M datasets, the ValGen ranking of algorithm snapshots seems to be only weakly correlated with the true meta-test ranking for all the meta-learning methods. In fact, the meta-val and meta-test rankings of the most useful snapshots can sometimes even be negatively correlated ($\rho \approx -0.12 < 0$ over all snapshots after epoch 30 in the training scenario shown in Figure 2(a)). These results show that on the OOD FSL benchmarks, snapshot selection using the pre-assigned validation tasks can sometimes be **unreliable/unable to identify a snapshot candidate with top-tier meta-test performance among all the snapshots.**

### 5.1.2 *Option 2*: Snapshot selection alternative using BaseGen.

Beyond using OOD validation tasks for snapshot selection, inspired by the domain generalization community [23, 18], we can alternatively also consider using the ID performance over the training task distribution for snapshot selection. Perhaps due to the lack of an ID evaluation setup in common FSL benchmarks, this possibility has not been widely considered. Enabled by our modifications of current FSL benchmarks, we can now evaluate the ID generalization performance (BaseGen) throughout training in addition to ValGen.

We plot how BaseGen and NovelGen progress for meta-learning methods trained on two different datasets in 2(c)(d). Here we see that on *tiered*-M, the BaseGen and NovelGen of PN both increase fairly consistently; thus picking the algorithm snapshot with the highest BaseGen performance (roughly the end-of-training snapshot) would also give close-to-best NovelGen. However, on *cifar*-M, after the learning rate drop at epoch 20, SVM's BaseGen keeps improving while NovelGen starts deteriorating. In this case, selecting snapshots according to the best BaseGen would pick a much worse snapshot than picking according to the best ValGen. (For concrete numbers of how much snapshot selection through BaseGen vs. ValGen could impact the chosen snapshot's NovelGen performance in each of these two cases, see Appendix F.3.) This ambiguity of whether ID or OOD Validation snapshot selection is better has also been documented in domain generalization, where Gulrajani and Lopez-Paz [18] find ID model selection can perform better in some settings while Koh et al. [23] find OOD validation model selection is better in others. Despite this ambiguity, we believe the commonly neglected **in-distribution (BaseGen) snapshot selection approach should be considered by users of OOD FSL benchmarks as a viable alternative** to the default ValGen selection approach in proper settings.

### 5.2 Inconsistencies in Meta-learning Method Performance Comparisons

After discussing concerns regarding OOD model selection for *a given meta-learning method*, we now analyze the reliability and consistency of conclusions drawn from comparing *different meta-learning methods*' OOD performance on these benchmarks. In particular, we focus on two cases:

*Inconsistency example 1*: **Limited number of novel classes in a single benchmark.** Since in OOD FSL we specifically care about the learned algorithms' ability to quickly learn many unseen concepts, we should not be satisfied with an algorithm performing well only on tasks constructed from a small number of pre-selected novel classes. However, for many widely-used FSL benchmarks

---

[§]$\rho \in [-1, 1]$, $\rho \approx 0$ means no correlation, while $\rho = 1/\rho = -1$ means exactly same/opposite rankings.

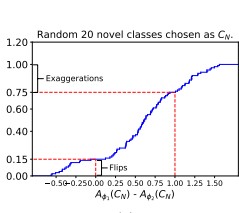

| | ε | $|\mathcal{C}_B|$ | $|\mathcal{C}_N|$ | $|\mathcal{C}_L|$ | CF | IE,0.5% |
|---|---|---|---|---|---|---|
| (IC) | 0.5% | 100 imgs/class | | | 3% | 1% |
| (i) | 0.5% | 64 | 20 | 120 | 15% | 25% |
| (ii) | 0.5% | 64 | 20 | 552 | 20% | 24% |
| (iii) | 1.0% | 64 | 20 | 552 | 10% | 12% |
| (iv) | 0.5% | 64 | 20 | 552 | 20% | 23% |
| (v) | 0.5% | 64 | 160 | 552 | 0% | 7% |
| (vi) | 0.5% | 351 | 20 | 552 | 9% | 15% |
| (vii) | 0.5% | 351 | 160 | 552 | 0% | 1% |

(Rows (IC)–(iii) are *mini*-M; rows (iv)–(vii) are *tiered*-M.)

| Dataset | ZAPPOS-OOD | | *mini*-OOD |
|---|---|---|---|
| Methods | 2w10s | 2w5s | 5w5s |
| PN | [1]80.51 ±0.13% | [1]74.67 ±0.14% | [3]76.22 ±0.14% |
| Ridge | [1]80.51 ±0.13% | [2]73.50 ±0.14% | [2]77.20 ±0.15% |
| SVM | [3]79.70 ±0.13% | [3]72.84 ±0.14% | [1]77.72 ±0.15% |
| FO-MAML | [4]72.89 ±0.14% | [4]68.12 ±0.15% | [4]75.96 ±0.17% |

(a)    (b)    (c)

Figure 3: In (a), we show the CDF plot of $A_{\phi_1}(\mathcal{C}_N) - A_{\phi_2}(\mathcal{C}_N)$ over 100 randomly chosen sets of $\mathcal{C}_N$ with 20 (out of total $|\mathcal{C}_L|=120$) novel classes each. In (b), for different values of true performance difference $\epsilon$ and values of underlying class size $|\mathcal{C}_L|$, training class size ($|\mathcal{C}_B|$), and evaluation class size ($|\mathcal{C}_N|$); we show the percentage of conclusion flips (CF) and improvement exaggerations (IE) with $\delta = 0.5\%$ computed over 100 evaluations. In (c), we demonstrate the inconsistencies in performance rankings for PN, SVM, RR, and FOMAML on two OOD benchmarks: Zappos-OOD and *mini*-OOD.

(*mini*ImageNet, CIFAR-FS, FC-100), only 20 novel classes are used for meta-testing. Ideally, even if we don't measure it, we would hope that our learned algorithm would also generalize to tasks made by other sets of classes different from the fixed small set of novel classes.

**Formal setup.** We suppose the existence of a much larger collection of classes $\mathcal{C}_L$, where the novel classes $\mathcal{C}_N$ used for meta-testing is a small random subset with each element class sampled uniformly and non-repeatedly from $\mathcal{C}_L$ and fixed thereafter for NovelGen evaluation. Ideally, our goal is to evaluate an algorithm snapshot $\phi$ on the task distribution $\mathbb{P}_{\mathcal{C}_L}$ (denote this performance by $A_\phi(\mathcal{C}_L)$), yet during evaluation we only have access to the novel classes in $\mathcal{C}_N$ and thus we can only compute the performance $A_\phi(\mathcal{C}_N)$. It is easy to see that when we randomize over different choices of $\mathcal{C}_N$, the expected performance over the sampled novel classes would match the true performance: $\mathbb{E}_{\mathcal{C}_N}[A_\phi(\mathcal{C}_N)] = A_\phi(\mathcal{C}_L)$. However, when using a single randomly sampled novel set, the estimator $A_\phi(\mathcal{C}_N)$ can have high variance (see Appendix F.6). Instead of relying on $A_\phi(\mathcal{C}_N)$ to directly estimate $A_\phi(\mathcal{C}_L)$, we ask a more relaxed question: for a pair of algorithms $A_{\phi_1}$ and $A_{\phi_2}$ (given by two meta-learning methods), if the true performance $A_{\phi_1}(\mathcal{C}_L) - A_{\phi_2}(\mathcal{C}_L) = \epsilon > 0$, how frequently will we observe an opposite conclusion *i.e.,* $\mathbb{P}(A_{\phi_1}(\mathcal{C}_N) < A_{\phi_2}(\mathcal{C}_N))$ over a randomly sampled $\mathcal{C}_N$ (we call this event *conclusion flip*)? Additionally, it is also possible that the observed performance difference on $\mathcal{C}_N$ is greater than the true difference $\epsilon$ by some amount $\delta > 0$. In this case, the NovelGen observation would make the algorithm snapshot $\phi_1$ look better than $\phi_2$ more than it actually is on $\mathcal{C}_L$. Thus we also ask what is the probability of $\mathbb{P}(A_{\phi_1}(\mathcal{C}_N) - A_{\phi_2}(\mathcal{C}_N) > \epsilon + \delta)$ and we call such events *improvement exaggerations*. To answer both these questions empirically, we first suggest some larger class sets $\mathcal{C}_L$ for *mini* and *tiered*. For both we select unused classes from ImageNet disjoint from the base and validation classes. For *tiered*, we use all the remaining $1000 - 351$ (base) $-97$ (novel) $= 552$ classes as $\mathcal{C}_L$ while for *mini*, we randomly choose 120 or 552 (to match $|\mathcal{C}_L|$ in *tiered*) unused classes. We fix these $\mathcal{C}_L$ choices in the following analysis.

**Checking the frequency of conclusion flips and exaggerations.** Figure 3(a) shows the empirical CDF of the performance differences $A_{\phi_1}(\mathcal{C}_N) - A_{\phi_2}(\mathcal{C}_N)$ computed over 100 randomly sampled size-20 novel class sets $\mathcal{C}_N$ for a fixed pair of PN and RR algorithm snapshots whose true performance difference over the larger 120 classes $\mathcal{C}_L$ is $\epsilon = 0.5\%$. In 15% of the cases the performance order is flipped from the true order, while in 25% of them improvements are exaggerated by more than 0.5% (total difference greater than 1%). Moreover, for some of the performance order flips, the observed performance difference can be quite negative $< -0.5\%$ thus significantly opposite to the true performance order. (Here for each run we evaluate both methods on 20,000 tasks sampled from $\mathbb{P}_{\mathcal{C}_N}$ in order to significantly reduce the randomness in estimating the true $A_{\phi_1}(\mathcal{C}_N), A_{\phi_2}(\mathcal{C}_N)$.)

**Comparison to supervised learning.** We check for the conclusion flip and improvement exaggeration frequency when only a random subset of the full test set (100 randomly drawn test images from each base class in *mini*-M) is used to compare two supervised learning image classification models with the same full test set performance difference of $\epsilon = 0.5$ (row (IC) in Table 3(b)). Here we see that **compared to supervised learning, the chances of getting an incorrect performance comparison** (row (i) in Table 3(b)) **is much higher for the meta-learning OOD FSL benchmarks** when evaluating only on 20 randomly chosen novel classes (as done in several FSL benchmarks).

**Larger $|\mathcal{C}_L|$ makes it even less reliable but larger $\epsilon$ helps.** If we were to care about an even larger set of underlying classes ($|\mathcal{C}_L| = 552$) despite still using only 20 random novel classes for evaluation

comparison, the conclusions are even less reliable (Table 3(b) (i) vs (ii)). On the other hand, we do see that the performance comparison becomes comparatively more consistent if the true performance difference $\epsilon$ is higher ($1\%$ in (iii) compared to $0.5\%$ in (ii)), despite that there still exists a statistically significant chance ($10\%$) of getting an opposite conclusion.

**OOD evaluations in current FSL benchmarks.** In practice, because **1)** we never specify exactly what and how big the underlying set of classes that we care about is, and **2)** some of the recent meta-learning methods (SVM vs PN on *cifar* in Table 2 of [25], R2-D2 vs GNN on *mini* in Table 1 of [4], FIX-ML [38]) sometimes only improve over the prior works by $< 1\%$, we believe researchers should **be aware of the possibility of getting a performance conclusion that is inconsistent** over a single randomly chosen and fixed set of 20 novel classes used by some of these benchmarks.

**Actionable suggestions.** Since the size of the unknown underlying larger class set $\mathcal{C}_L$ and the true performance difference $\epsilon$ might not be something one can directly control when designing the OOD benchmark, we now discuss two actionable choices that can reduce the chances of conclusion flips:

i) **Use more novel classes in the evaluation:** By comparing (iv) vs (v) and (vi) vs (vii) in Table 3, we see that the frequency of conclusion flips and improvement exaggerations are **much** lower when 160 novel classes are used as opposed to 20 when $|\mathcal{C}_L|$ is the same.
ii) **Train on more base classes:** The *tiered* dataset has more base classes (351 compared to 64 for *mini*) to train on. When comparing PN and RR snapshots trained on a modified version of *tiered* with fewer (randomly sampled 64 out of 351 to match *mini*) base classes, we see that the CF frequency is twice as high compared to when 351 base classes are used (Table 3(b)(iv) vs (vi)).

Based on these two trends, for more reliable comparisons of meta-learning methods' OOD performance we suggest using datasets like *tiered*ImageNet and MetaDataset (both with much larger set of base and novel classes) in addition to the smaller benchmarks like *mini*ImageNet, CIFAR-FS, and FC-100, which some recent works [e.g., 30, 4] still solely rely upon.

*Inconsistency example 2*: **Inconsistency across multiple OOD FSL benchmarks.** Unlike the ID scenario where the training and test task distribution are the same, the similarity between training and test distributions in the OOD FSL benchmarks can vary significantly. Ideally, we want a meta-learning method to be consistently better on multiple OOD benchmarks with different type/degree of distribution shifts. Since Ren et al. [35] originally use the Zappos dataset for OOD evaluation, we also perform a similar evaluation on new attribute pairs based on their setup. At test time, we use an attribute set $\mathcal{A}'$ disjoint from the one used in the Zappos-ID setup $\mathcal{A}$, and sample attribute pairs from $\mathcal{A}'$ only. This induces a test task distribution $\mathbb{P}_{\mathcal{A}'}$ different from the training task distribution $\mathbb{P}_{\mathcal{A}}$. We evaluate different meta-learning methods on these Zappos-OOD tasks to see if the performance order is consistent with other OOD FSL benchmarks (Table 3(c)). Here we see that despite SVM outperforming RR and PN on *mini* NovelGen, the performance order of these three methods are completely flipped on Zappos-OOD. Similar observations can be made from TADAM underperforming PN in Table 2 of [35] despite TADAM being shown to outperform PN on the other more commonly-used FSL benchmarks. *This inconsistency over different types of OOD FSL benchmarks is in stark contrast to the consistency of performance rankings over the 6 different ID benchmarks* (FEMNIST, Zappos-ID, and the BaseGen results of the 4 current FSL benchmarks (Section 4)). Based on these findings, we caution meta-learning researchers to **be aware of such conclusion inconsistencies over different OOD FSL scenarios** and **reason carefully about the generality of their empirical findings** when using only specific types of OOD datasets.

## 6   Conclusion

In this paper, we categorize meta few-shot learning evaluation into two settings: in-distribution (ID) and out-of-distribution (OOD). After explaining why common FSL benchmarks reflect OOD evaluation, we identify realistic needs for ID FSL evaluation and provide new benchmarks as well as suggestions on how to modify existing OOD FSL benchmarks to allow for ID evaluation. Through experiments performed on these ID benchmarks, we demonstrate a surprising phenomenon that many meta-learning methods/training techniques improve OOD performance while sacrificing ID performance. Beyond this, through quantitative analyses, we show that even in the OOD scenario, current FSL benchmarks may present subtle challenges with both model selection for a given meta-learning method and reliable performance comparisons of different methods. For these concerns, we provide initial suggestions and alternatives with the hope of alleviating these issues. Overall, we aim to raise awareness about the dichotomy of FSL evaluation and to motivate the meta-learning community to collectively reason about ways to improve both ID and OOD methodology and evaluation.

**Acknowledgements.** This work was supported in part by the National Science Foundation Grant IIS1838017, a Google Faculty Award, a Facebook Faculty Award, and the CONIX Research Center. Any opinions, findings, and conclusions or recommendations expressed in this material are those of the author(s) and do not necessarily reflect the NSF or any other funding agency.

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
