# Appendix

## Appendix Outline

## A  Assumption (ID) and Evaluation (OOD) Mismatch Examples

To illustrate the mismatch between meta-learning theory/methodology and evaluation (Section 2), below are some examples of works that motivate commonly-used meta-learning methods in the in-distribution setting, but largely evaluate empirical performance on OOD FSL benchmarks. For convenience, we provide exact lines from the original works that refer to the ID scenario. Our aim is not to draw attention to these works specifically, but to highlight the ubiquity of the divide between theory and practice in current meta-learning literature.

- Lee et al. [25] (Section 3.1): "It is often **assumed that the training and test set are sampled from the same distribution** and the domain is mapped to a feature space using an embedding model $f_\phi$ parameterized by $\phi$ ".
- Rajeswaran et al. [31] (Section 2.1): "... a collection of **meta-training** tasks $\{\mathcal{T}_i\}_{i=1}^M$ **drawn from** $P(\mathcal{T})$ ... At **meta-test** (deployment) time, when presented with a dataset $\mathcal{D}_j^{\mathrm{tr}}$ corresponding to a **new task** $\mathcal{T}_j \sim P(\mathcal{T})$." Notice that the training and test tasks are all sampled from the same task distribution $P(\mathcal{T})$.
- Finn et al. [16] (Section 3): "To do so, meta-learning algorithms require a set of **meta-training and meta-testing tasks drawn from some distribution** $p(\mathcal{T})$. The key assumption of learning-to-learn is that the tasks in this distribution share common ... ".

## B  Formal Analysis on Why Current FSL Benchmarks Are OOD

In this section we provide formal arguments for the informal statements in Section 3, which explain why it is improbable for the train and test tasks in the current FSL benchmarks to be *iid* sampled from the same underlying distribution.

**Formal Setup.** If we believe that both train and test tasks in current FSL benchmarks are sampled from the same underlying task distribution, then this shared task distribution (where each task is specified by a class tuple) must cover a larger set of underlying classes $\mathcal{C}_L$ which would contain both the base classes and novel classes as subsets $\mathcal{C}_L \supseteq (\mathcal{C}_B \cup \mathcal{C}_N)$. For convenience, we represent the classes in this set with $\mathcal{C}_L := \{1, \ldots, L\}$ where the task distribution (from which the train and test tasks are *iid* sampled) is induced by a probabilistic distribution over $n-$way non-repeating tuples $\boldsymbol{c} := (c_1, \ldots, c_n) \in [\mathcal{C}_L^n]$, denoted by $\mathbb{P}_L(\boldsymbol{c})$. To sample a task from this larger task distribution, we sample $\boldsymbol{c} \sim \mathbb{P}_L$ and take the corresponding task $\mathcal{T}_{\boldsymbol{c}}$. Notice that this task distribution can be more general than $\mathbb{P}_{\mathcal{C}_N}$ or $\mathbb{P}_{\mathcal{C}_B}$, as $\mathbb{P}_L(\boldsymbol{c})$ does not have to be a uniform distribution over all possible class tuples.

**Definition 1** (Probability of observing a class in a single draw). *The indicator event of observing a class $i$ anywhere in a randomly drawn class tuple $\boldsymbol{c} \sim \mathbb{P}_L$ can be represented by $\sum_{j=1}^n \mathbb{1}(c_j = i)$, since it is impossible to observe the same class more than once in the same tuple. We denote the probability of this event by $p_i := \mathbb{P}\left(\left(\sum_{j=1}^n \mathbb{1}(c_j = i)\right) = 1\right) = \sum_{j=1}^n \mathbb{P}(\mathbb{1}(c_j = i) = 1) = \sum_{j=1}^n \mathbb{P}(c_j = i)$.*

**Lemma 1.** *The sum of the probability of observing a class $i$ in a single class tuple draw over all the classes $i \in \mathcal{C}_L$ is equal to $n$, i.e., $\sum_{i=1}^L p_i = n$.*

*Proof of Lemma 1.* We know that $\sum_{i=1}^{L} p_i = \sum_{i=1}^{L} \sum_{j=1}^{n} \mathbb{P}(c_j = i)$ by simply plugging in the definition of $p_i$. Since $\sum_{i=1}^{L} \mathbb{P}(c_j = i) = 1$, by exchanging the summations we get, $\sum_{i=1}^{L} p_i = \sum_{j=1}^{n} \sum_{i=1}^{L} \mathbb{P}(c_j = i) = \sum_{j=1}^{n} [1] = n$. $\qquad\square$

**Assumption 1** (Every class must have nonzero probability to be sampled). *To avoid degeneracy, we assume that each class has a minimum non-zero probability of being sampled in a class tuple: $\forall i \in \{1, \ldots, L\}$, $1 \geq p_i \geq \frac{\gamma n}{L}$, where $\gamma \in (0, 1]$. Notice $\gamma$ is strictly greater than 0 to avoid the degenerate case where a class would almost surely never be sampled in any class tuple. If there exists such a class, then we can prune the set $\mathcal{C}_L$ accordingly and use the pruned set (which now has every class with nonzero probability) as our new $\mathcal{C}_L$.*

**Remark.** Note that the task distribution induced by the probability values $\{p_i\}_{i=1}^{L}$ in Assumption 1 is a relaxed form of the uniform distribution $\text{Unif}(\{\mathcal{T}_c := c \in [\mathcal{C}_L^n]\})$ over all non-repeating class tuples spawned by $\mathcal{C}_L$. This case can be recovered by setting $p_i = \frac{n}{L}$, $\forall i \in [L]$.

Suppose there are $N$ total *iid* random draws $\{c^{(k)}\}_{k=1}^{N}$ of class tuples from $\mathbb{P}_L$ (every $c^{(k)} \in [\mathcal{C}_L^n]$), then the event of observing a class $i$ in any of these $N$ class tuple draws is exactly the complement of the event that the class does not appear in any of these tuples.

**Definition 2** (Observing a class at least once in $N$ draws). *We denote the indicator random variable of observing a class $i$ in any of the $N$ draws by*

$$X_{i,N} := 1 - \mathbb{1}\left(\left(\sum_{k=1}^{N} \mathbb{1}(i \in c^{(k)})\right) = 0\right) \in \{0, 1\}. \tag{1}$$

*Then we have $\mathbb{E}[X_{i,N}] = \mathbb{P}(X_{i,N} = 1) = 1 - (1 - p_i)^N$. We denote the random variable representing the total number of unique classes observed in $N$ draws as $Z$, which can be expressed by*

$$Z = \sum_{i=1}^{L} X_{i,N}. \tag{2}$$

**Remark.** We note that the total number of unique classes seen ($Z$) in $N$ *iid* draws **1)** must have at least $n$ classes (even after a single class tuple is sampled, there would already be $n$ different classes seen) and cannot be greater than the total number of classes possible, *i.e.*, $Z \in [n, L]$, and **2)** cannot be greater than the total number of (possibly overlapping) classes drawn, *i.e.*, $Z \leq nN$.

**Lemma 2.** *For notational convenience, let $q_i := 1 - p_i$. Then, by Equation (1) and Assumption 1 we have:*

*(a)* $\mathbb{E}[Z] = L - \sum_{i=1}^{L} (1 - p_i)^N = L - \sum_{i=1}^{L} q_i^N,$ $\qquad\qquad$ (3)

*(b)* *For $\{q_i\}_{i=1}^{L}$, $0 \leq q_i \leq 1 - \frac{\gamma n}{L}$ and $\sum_{i=1}^{L} q_i = L - n$.* $\qquad$ (4)

Now that we have set up the problem formulation, we provide Theorems 1, 2 describing properties of $\mathbb{E}[Z]$ and $\mathbb{V}[Z]$ which we will use to analyze the dichotomy described in the main paper (Section 3).

## B.1 Lower Bound on $\mathbb{E}[Z]$

To achieve a lower bound of $\mathbb{E}[Z]$, we need to analyze the worst case class tuple distribution that makes the value $L - \sum_{i=1}^{L} q_i^N$ as small as possible. This amounts to maximizing the value of $\sum_{i=1}^{L} q_i^N$ under the constraints for $\{q_i\}_{i=1}^{L}$ described in Lemma 2. We present an upper bound for this constrained maximization objective below.

**Theorem 1** (Lower bound on $\mathbb{E}[Z]$)**.** *The optimal value of the following constrained optimization problem in* (5) *is upper bounded by* $L\left(1 - \frac{\gamma n}{L}\right)^N$.

$$\max_{\{q_i\}_{i=1}^L} \quad \sum_{i=1}^L q_i^N \tag{5}$$

$$\text{subject to} \quad 0 \leq q_i \leq 1 - \frac{\gamma n}{L}, \ \forall i \in [L]$$

$$\sum_{i=1}^L q_i = L - n$$

*As a result, for $Z$ defined in* (2), *directly applying* (3) *we get* $\mathbb{E}[Z] \geq L\left(1 - (1 - \frac{\gamma n}{L})^N\right)$.

To prove Theorem 1, we first provide a lemma describing the structure of the solution to the optimization problem in (5).

**Lemma 3** (Structure of the optimal solution to (5))**.** *The optimal solution to optimization Objective* (5) *has the following form: out of the $L$ variables $\{q_i\}_{i=1}^L$, $K$ of them have value $1 - \frac{\gamma n}{L}$, $(L - K - 1)$ of them have value $0$, and the last remaining variable has the value $(L - n) - K(1 - \frac{\gamma n}{L})$, which must still be in the range of $\left[0, 1 - \frac{\gamma n}{L}\right]$. This directly implies that the integer $K$ must be satisfy $\frac{L^2 - nL}{L - \gamma n} - 1 \leq K \leq \frac{L^2 - nL}{L - \gamma n}$.*

*Proof of Lemma 3.* Let us denote the optimal solution to Objective (5) by $(q_1^*, \ldots, q_n^*)$. Suppose that there exists a pair $q_k^*, q_j^*$, $k \neq j$, such that neither of them equals $0$ or $1 - \frac{\gamma n}{L}$. Then by changing the values of $q_k^*, q_j^*$ to be either $(1 - \frac{\gamma n}{L}, q_k^* + q_j^* - 1 + \frac{\gamma n}{L})$ or $(0, q_k^* + q_j^*)$, the new $q$ tuple would still be feasible while the value of Objective 5 would strictly improve because the function $q_k^N + q_j^N$ is strongly convex over $\mathbb{R}_{++}^2$. (Recall that a convex function over a closed interval can only take maximum value at either one of its two endpoints.) As a result, there can be no more than a single $q_i^*$ in the optimal solution $(q_1^*, \ldots, q_L^*)$ that has a value of neither $0$ nor $1 - \frac{\gamma n}{L}$. Now, we denote the total number of $q_i$'s in the optimal solution that has the value of $1 - \frac{\gamma n}{L}$ by $K \in \mathbb{Z}$, then there must be at least $L - K - 1$ values of $0$, with the remaining term $(L - n) - K(1 - \frac{\gamma n}{L}) \in \left[0, 1 - \frac{\gamma n}{L}\right]$. Manipulating this inequality of $K$ gives us the feasible range of $K$, $\frac{L^2 - nL}{L - \gamma n} - 1 \leq K \leq \frac{L^2 - nL}{L - \gamma n}$. $\square$

*Proof of Theorem 1.* We know from Lemma 3 that the optimal solution to the constrained maximization problem in Theorem 1 is given by the optimal solution to the reduced objective below.

$$\max_{K \in \{0, \ldots, L\}} \quad K \cdot \left(1 - \frac{\gamma n}{L}\right)^N + \left[L - n - K \cdot \left(1 - \frac{\gamma n}{L}\right)\right]^N \tag{6}$$

$$\text{subject to} \quad \frac{L^2 - nL}{L - \gamma n} - 1 \leq K \leq \frac{L^2 - nL}{L - \gamma n} \tag{7}$$

Let the optimal value of $K$ (minimum value if multiple are optimal) in the above optimization problem be $K^*$. Then the optimal value of the above is upper bounded by:

$$K^* \cdot \left(1 - \frac{\gamma n}{L}\right)^N + \left[L - n - K^* \cdot \left(1 - \frac{\gamma n}{L}\right)\right]^N$$

$$\leq (K^* + 1) \cdot \left(1 - \frac{\gamma n}{L}\right)^N$$

$$\leq L \cdot \left(1 - \frac{\gamma n}{L}\right)^N$$

As a result, directly applying (3), we have:

$$\mathbb{E}[Z] \geq L \cdot \left[1 - \left(1 - \frac{\gamma n}{L}\right)^N\right]. \tag{8}$$

$\square$

## B.2 Upper bound on $\mathbb{V}[Z]$

We apply the Efron-Stein inequality to obtain the upper bound of the variance of $Z$, which we state here for convenience.

**Lemma 4** (Efron-Stein's inequality). *Let $S : \mathcal{Y}^N \to \mathbb{R}$ be a measurable function that is permutation invariant. Let the random variable $U$ be given by $U = S(Y_1, \dots, Y_N)$, where $(Y_1, \dots, Y_N)$ is a random vector of $N$ independent random variables in $\mathcal{Y}^N$. Then, we have:*

$$\mathbb{V}[U] \leq \frac{1}{2} \sum_{i=1}^{N} \mathbb{E}(U - U_i')^2, \tag{9}$$

*where $U_i' = S(Y_1, ..., Y_i', ..., Y_N)$, and $\forall i \in [N]$, $Y_i$ and $Y_i'$ are drawn iid from the same distribution.*

**Theorem 2** (Upper bound on $\mathbb{V}[Z]$). *For $Z$ defined in (2), the variance $\mathbb{V}[Z] \leq \frac{1}{2}n^2 N$.*

*Proof of Theorem 2.* By directly applying Lemma 4 on the permutation invariant measurable function $S : [\mathcal{C}_L^n]^N \to \mathbb{R}$, where $Z = S(\boldsymbol{c}^{(1)}, \dots, \boldsymbol{c}^{(N)}) := \sum_{i=1}^{L} X_{i,N}$, we have the variance $\mathbb{V}[Z] \leq \frac{1}{2}\sum_{i=1}^{N} \mathbb{E}(Z - Z_i')^2 \leq \frac{1}{2}\sum_{i=1}^{N} n^2 = \frac{1}{2}n^2 N$. Note that the last inequality holds because when we swap out one observed class tuple $\boldsymbol{c}^{(k)}$ (among the $N$ total) with a different one $\boldsymbol{c}^{(k)'}$ (to get $Z_i'$), the total number of unique classes we observe can change by at most $n$, *i.e.*, $|Z - Z_i'| \leq n$. $\qquad\square$

Using Theorems 1, 2, we now show that it is extremely unlikely for the commonly used FSL benchmarks, which have a (relatively) small number of train classes and disjoint train/test classes, to have their training and test tasks sampled *iid* from the same underlying task distribution. We break this analysis into the dichotomy presented in the main paper (Section 3).

## B.3 When $|\mathcal{C}_L|$ is Small ($L = \mathcal{O}(nN)$)

**Definition 3.** *Let $Z_{\mathrm{tr}}$ be the total number of unique classes observed in $N_{\mathrm{tr}}$ iid drawn train class tuples (tasks) from $\mathbb{P}_L$ and similarly let $Z_{\mathrm{te}}$ be the total number of unique classes observed in $N_{\mathrm{te}}$ iid drawn test class tuples (tasks) from $\mathbb{P}_L$. Furthermore, denote the set of indices of unique train classes by $\mathcal{I}_{\mathrm{tr}} := \{j : X_{j,N_{\mathrm{tr}}} = 1, j \in [N_{\mathrm{tr}}]\}$ and the set of unique test classes be $\mathcal{I}_{\mathrm{te}} := \{j : X_{j,N_{\mathrm{te}}} = 1, j \in [N_{\mathrm{te}}]\}$. Under this notation, the probability of observing disjoint sets of train and test classes among the $N_{\mathrm{tr}}$ and $N_{\mathrm{te}}$ randomly drawn train and test class tuples can be denoted by $\mathbb{P}(\mathcal{I}_{\mathrm{tr}} \cap \mathcal{I}_{\mathrm{te}} = \phi)$.*

**Theorem 3** (Upper bounding the probability of having disjoint train, test classes). *$\mathbb{P}(Z_{\mathrm{tr}} + Z_{\mathrm{te}} \leq L) \leq 4\left(1 - \frac{\gamma n}{L}\right)^{min(N_{\mathrm{tr}}, N_{\mathrm{te}})}$. As a result, $\mathbb{P}(\mathcal{I}_{\mathrm{tr}} \cap \mathcal{I}_{\mathrm{te}} = \phi) \leq \mathbb{P}(Z_{\mathrm{tr}} + Z_{\mathrm{te}} \leq L) \leq 4\left(1 - \frac{\gamma n}{L}\right)^{min(N_{\mathrm{tr}}, N_{\mathrm{te}})}$.*

*Proof of Theorem 3.* Since the random variable $L - Z_{\mathrm{tr}} \geq 0$, using Markov's inequality we have:

$$\mathbb{P}\left(Z_{\mathrm{tr}} \leq \frac{L}{2}\right) = \mathbb{P}\left(L - Z_{\mathrm{tr}} \geq L - \frac{L}{2}\right) \tag{10}$$

$$\leq \frac{L - \mathbb{E}[Z_{\mathrm{tr}}]}{(L - \frac{L}{2})}$$

$$\leq \frac{2(L - \mathbb{E}[Z_{\mathrm{tr}}])}{L}$$

$$\leq 2\left(1 - \frac{\gamma n}{L}\right)^{N_{\mathrm{tr}}}. \qquad \text{(using Theorem 1)} \tag{11}$$

Similarly, since $L - Z_{\mathrm{te}} \geq 0$, we have

$$\mathbb{P}\left(Z_{\mathrm{te}} \leq \frac{L}{2}\right) \leq 2\left(1 - \frac{\gamma n}{L}\right)^{N_{\mathrm{te}}}. \tag{12}$$

Since $\mathbb{P}\left(Z_{\mathrm{tr}} + Z_{\mathrm{te}} \leq L\right) \leq \mathbb{P}\left(\left(Z_{\mathrm{tr}} \leq \frac{L}{2}\right) \cup \left(Z_{\mathrm{te}} \leq \frac{L}{2}\right)\right)$, applying the union bound yields:

$$\mathbb{P}\left(Z_{\mathrm{te}} + Z_{\mathrm{te}} \leq L\right) \leq 4\left(1 - \frac{\gamma n}{L}\right)^{\min(N_{\mathrm{tr}}, N_{\mathrm{te}})}. \tag{13}$$

When $Z_{\mathrm{tr}} + Z_{\mathrm{te}} > L$, by pigeonhole principle, the two sets of class indices $\mathcal{I}_{\mathrm{tr}}$ and $\mathcal{I}_{\mathrm{te}}$ must have non-empty intersection, *i.e.*, $\mathcal{I}_{\mathrm{tr}} \cap \mathcal{I}_{\mathrm{te}} \neq \phi$. Taking the contra-positive of this claim, we see that $\mathcal{I}_{\mathrm{tr}} \cap \mathcal{I}_{\mathrm{te}} = \phi$ implies the event $Z_{\mathrm{tr}} + Z_{\mathrm{te}} \leq L$. As a result, we have $\mathbb{P}(\mathcal{I}_{\mathrm{tr}} \cap \mathcal{I}_{\mathrm{te}} = \phi) \leq \mathbb{P}(Z_{\mathrm{tr}} + Z_{\mathrm{te}} \leq L) \leq 4\left(1 - \frac{\gamma n}{L}\right)^{\min(N_{\mathrm{tr}}, N_{\mathrm{te}})}$. $\qquad \square$

**Corollary 1.** *If enough samples are observed i.e., if* $\min(N_{\mathrm{tr}}, N_{\mathrm{te}}) \geq \frac{\ln(4/\rho)L}{\gamma n}$*, then the probability of having no training and test classes intersection is upper bounded by* $\mathbb{P}(\mathcal{I}_{\mathrm{tr}} \cap \mathcal{I}_{\mathrm{te}} = \phi) \leq \rho$.

*Proof of Corollary 1.* By logarithm inequality $\ln(1 + x) \geq \frac{x}{1+x}$, we have $\ln(L/(L - \gamma n)) \geq \gamma n/L$. Taking the reciprocal of the two sides, we have $\frac{L}{\gamma n} \geq \frac{1}{\ln(L/(L-\gamma n))}$. As a result, $\min(N_{\mathrm{tr}}, N_{\mathrm{te}}) \geq \frac{\ln(4/\rho)L}{\gamma n} \geq \frac{\ln(4/\rho)}{\ln(L/(L-\gamma n))} = \frac{\ln(\rho/4)}{\ln(1 - \frac{\gamma n}{L})} = \log_{(1-\frac{\gamma n}{L})}\left(\frac{\rho}{4}\right)$, where the last step uses the change of basis equality of logarithm. Thus, by $\min(N_{\mathrm{tr}}, N_{\mathrm{te}}) \geq \log_{(1-\frac{\gamma n}{L})}\left(\frac{\rho}{4}\right)$, we have

$$\begin{aligned}
&\mathbb{P}(\mathcal{I}_{\mathrm{tr}} \cap \mathcal{I}_{\mathrm{te}} = \phi) \\
\leq\ & \mathbb{P}(Z_{\mathrm{tr}} + Z_{\mathrm{te}} \leq L) \\
\leq\ & 4\left(1 - \frac{\gamma n}{L}\right)^{\min(N_{\mathrm{tr}}, N_{\mathrm{te}})} \\
\leq\ & 4\left(1 - \frac{\gamma n}{L}\right)^{\log_{(1-\frac{\gamma n}{L})}\left(\frac{\rho}{4}\right)} \\
=\ & 4 \cdot \frac{\rho}{4} \\
=\ & \rho.
\end{aligned}$$

$\square$

**Remark.** From Corollary 1, we see that when the number of tasks sampled is larger than a multiple of the number of underlying classes (for example when $\rho = 0.01$, $\gamma = 0.5$, $n = 5$, $\min(N_{\mathrm{tr}}, N_{\mathrm{te}}) \geq \frac{\ln(4/\rho)L}{\gamma n} \approx 2.39L$), and equivalently, $L = \mathcal{O}(nN)$, the probability of having no training and test task classes intersecting is upper bounded by $\mathbb{P}(\mathcal{I}_{\mathrm{tr}} \cap \mathcal{I}_{\mathrm{te}} = \phi) \leq \rho$ (in our example, the probability is upper bounded by $\rho = 0.01$, which is a statistically rare event). In summary, in this case, we show that when $L \leq cnN$ for some small constant $c$, the probability of having no intersection between the training and test task classes is extremely small because it is very likely that the training tasks and test tasks would each cover a majority ($\geq 50\%$) of the entire set of classes.

### B.4 When $|\mathcal{C}_L|$ is Large ($L = \Omega(nN)$)

In this alternate case, we analyze the scenario where the underlying set of classes is larger than the total number of tasks we sampled, for which we make the following assumption:

**Assumption 2.** $L \geq nN$, *i.e., even if we observe all the classes in the randomly drawn $N$ tuples to be distinct, we still would not exhaust the much larger underlying set $\mathcal{C}_L$. In this setting,* $L = \Omega(nN)$.

**Corollary 2.** *By Assumption 2 and the Bernoulli inequality* $(1 + x)^r \leq 1 + \frac{rx}{1-(r-1)x}$, $x \in \left(-1, \frac{1}{r-1}\right), r > 1$*, substituting* $x = \frac{-\gamma n}{L}$*,* $r = N$*, we can further lower bound the RHS of Equation* (8) *in Theorem 1:*

$$\mathbb{E}[Z] \geq \frac{\gamma n N}{(1 + \gamma)} \tag{14}$$

**Theorem 4** (Unlikely to observe only a small number of unique classes). *For* $\eta \in (0, 1)$*, the probability of observing totally fewer than* $\frac{\eta \gamma n N}{1+\gamma}$ *classes in $N$ iid class tuple samples from $\mathbb{P}_L$ is at most* $\frac{(1+\gamma)^2}{2(1-\eta)^2 \gamma^2 N}$.

*Proof.*

$$\mathbb{P}\left(Z \leq \frac{\eta\gamma nN}{(1+\gamma)}\right) \tag{15}$$

$$\leq \mathbb{P}(Z \leq \eta\,\mathbb{E}[Z])$$

$$= \mathbb{P}(Z \leq \mathbb{E}[Z] - (1-\eta)\,\mathbb{E}[Z])$$

$$\leq \mathbb{P}(|Z - \mathbb{E}[Z]| \geq (1-\eta)\,\mathbb{E}[Z])$$

$$\leq \frac{\mathbb{V}[Z]}{((1-\eta)\,\mathbb{E}[Z])^2} \qquad \text{(by Chebyshev's inequality)}$$

$$\leq \frac{\frac{1}{2}n^2 N}{(1-\eta)^2\,\mathbb{E}[Z]^2} \qquad \text{(using Theorem 2)}$$

$$\leq \frac{\frac{1}{2}n^2 N}{(1-\eta)^2\left(\frac{\gamma nN}{(1+\gamma)}\right)^2} \qquad \text{(using Corollary 2)}$$

$$= \frac{(1+\gamma)^2}{2(1-\eta)^2\gamma^2 N} \tag{16}$$

$\square$

In summary, in this case, when $L \geq nN$, the probability of observing only a small fraction of $nN$ classes in $N$ tuple draws, scales with $1/N$. Because in practice a very large number of training tasks are used ($N \geq 10^5$), the probability of only observing fewer than hundreds of classes ($Z \leq 10^3$) in $N$ class tuple samples would be extremely small. This means that we shouldn't treat the large number of tasks used during meta-training as being sampled *iid* from an underlying task distribution under the assumption that the number of task samples hasn't exceeded the total number of underlying classes.

## B.5 Concluding Remarks

In the first part of the dichotomy, we show using Theorem 3 that when the number of underlying classes is smaller than $nN$, it is highly unlikely for the train and test classes to be completely disjoint.

In the second part of the dichotomy, we show using Theorem 4 that when the number of underlying classes is larger than $nN$, it is unlikely to observe only a few (very small fraction of $nN$) unique train classes — in fact the number of unique train classes observed would roughly speaking scale linearly with the number of task samples $N$.

**Conclusion on why current FSL benchmarks target OOD.** Note that in current FSL benchmarks **i)** there is no overlap of classes observed in the train and test tasks; and **ii)** the number of train (base) classes observed (*e.g.,* 64 for *mini*) is much smaller than the total number of train tasks (*e.g.,* $\approx 10^6$ for *mini*). Thus, the two sides of the dichotomy (above) when taken together leads us to reject the hypothesis/assumption of *iid* sampled train and test tasks in the current FSL benchmarks.

# C  Overview of Notations for $(S, Q)$ Sampling in ID and OOD Benchmarks

Table 2: An overview of the notations used to describe each of the two steps: i) sampling the task from the training/test task distribution, and ii) sampling (S,Q) pair from the task; for the OOD benchmarks (*mini*, *cifar*, FC, *tiered*, Zappos-OOD) and ID benchmarks (FEMNIST, Zappos-ID).

| Benchmark / Steps | | Step 1: $\mathcal{T} \sim \mathbb{P}(\mathcal{T})$ | Step 2: $(S, Q) \sim \mathcal{T}$ |
|---|---|---|---|
| *mini*, *cifar*, FC, | Train | $\mathcal{T}_{\boldsymbol{c}_B} \sim \mathbb{P}_{\mathcal{C}_B} := \mathrm{Unif}(\{\mathcal{T}_{\boldsymbol{c}_B} : \boldsymbol{c}_B \in [\mathcal{C}_B^n]\})$ | $S, Q \sim \mathcal{T}_{\boldsymbol{c}_B}$ |
| *tiered* (OOD) | Test | $\mathcal{T}_{\boldsymbol{c}_N} \sim \mathbb{P}_{\mathcal{C}_N} := \mathrm{Unif}(\{\mathcal{T}_{\boldsymbol{c}_N} : \boldsymbol{c}_N \in [\mathcal{C}_N^n]\})$ | $S, Q \sim \mathcal{T}_{\boldsymbol{c}_N}$ |
| Zappos-OOD | Train | $\mathcal{T}_{\boldsymbol{a}} \sim \mathbb{P}_{\mathcal{A}}(\mathcal{T}) = \mathrm{Unif}(\{\mathcal{T}_{\boldsymbol{a}} : \boldsymbol{a} \in [\mathcal{A}^2]\})$ | $S, Q \sim \mathcal{T}_{\boldsymbol{a}}$ |
| (OOD) | Test | $\mathcal{T}_{\boldsymbol{a}'} \sim \mathbb{P}_{\mathcal{A}'}(\mathcal{T}) = \mathrm{Unif}(\{\mathcal{T}_{\boldsymbol{a}'} : \boldsymbol{a}' \in [\mathcal{A}'^2]\})$ | $S, Q \sim \mathcal{T}_{\boldsymbol{a}'}$ |
| FEMNIST | Train | $\mathcal{T}_{\mathtt{id}} \sim \mathbb{P}(\mathtt{id})$ | $S, Q \sim \mathcal{T}_{\mathtt{id}}$ |
| (ID) | Test | $\mathcal{T}_{\mathtt{id}} \sim \mathbb{P}(\mathtt{id})$ | $S, Q \sim \mathcal{T}_{\mathtt{id}}$ |
| Zappos-ID | Train | $\mathcal{T}_{\boldsymbol{a}} \sim \mathbb{P}_{\mathcal{A}}(\mathcal{T}) = \mathrm{Unif}(\{\mathcal{T}_{\boldsymbol{a}} : \boldsymbol{a} \in [\mathcal{A}^2]\})$ | $S, Q \sim \mathcal{T}_{\boldsymbol{a}}$ |
| (ID) | Test | $\mathcal{T}_{\boldsymbol{a}} \sim \mathbb{P}_{\mathcal{A}}(\mathcal{T}) = \mathrm{Unif}(\{\mathcal{T}_{\boldsymbol{a}} : \boldsymbol{a} \in [\mathcal{A}^2]\})$ | $S, Q \sim \mathcal{T}_{\boldsymbol{a}}$ |

# D  Dataset Preprocessing and Hyperparameter Details

Here we first provide some details on the logic used to construct the ID benchmark Zappos-ID and its OOD counterpart Zappos-OOD. We then list the set of hyperparameter configurations used to train the meta-learning methods PN, RR, SVM, FOMAML and the supervised learning baselines MB, SB and IC on each of the benchmarks in the paper.

## D.1  Zappos Preprocessing

Recall that the Zappos dataset is motivated through an online shopping recommendation problem, where each task is a binary classification of shoe images into an attribute context. Every online user is represented by such a task, where the user's preference for shoes is specified by the corresponding shoe attribute context. We consider a simplified setting where we fix a set of universal shoe attributes $\mathcal{A}$ and each user's preference is specified exactly by a pair of attributes $\boldsymbol{a} = (a_1, a_2) \in \mathcal{A}^2, \ a_1 \neq a_2$. The Zappos-ID and Zappos-OOD FSL benchmarks we use are derived from the UT Zappos50k corpus which consists of 50,025 shoe images each annotated with a list of attributes the shoe possess.

**Attributes Selection.** We limit the subset of attributes we consider to the 78 considered by Ren et al. [35] (Table 7). Recall that the task distribution we consider is the uniform distribution $\mathcal{T}_{\boldsymbol{a}} \sim \mathbb{P}_{\mathcal{A}}(\mathcal{T}) = \mathrm{Unif}(\{\mathcal{T}_{\boldsymbol{a}} : \boldsymbol{a} \in [\mathcal{A}^2]\})$ (Table 2) over all non-repeating attribute pairs in $\mathcal{A}$. In order to ensure that each attribute pair in $\mathcal{A}$ has at least 20 shoes carrying both the attributes (feasible pairs), we only consider the uniform distribution over such feasible attribute pairs. Thus, we reduce the original set of 78 attributes to 66, since 12 of the attributes where found to be infeasible with every other attribute in the original set.

**Determining $\mathcal{A}, \mathcal{A}'$.** For the Zappos-ID benchmark we use the set of attributes specified by $\mathcal{A}$ (of size 36) to *iid* sample 1000 (or 50) train and 25000 test tasks. On the other hand, as mentioned in Section 5 we use a disjoint set of attributes $\mathcal{A}'$ (of size 30) to sample 25000 test tasks for Zappos-OOD (see Table 3 for the exact sets). To determine this partition, we first consider a graph of 66 nodes, where each node represents an attribute and an undirected edge between a pair of attribute node is weighted by the number of shoe images (in the corpus) that have both attributes. Using spectral clustering, we find an approximate min-cut bipartition of this graph. In other words, we partition the entire set of attributes into two subsets in a way that reduces the number of images which carry pairs of attributes that are not in the same subset. This graph partition gives us the split of a 36-attribute set ($\mathcal{A}$) and a 30-attribute set ($\mathcal{A}'$).

Table 3: We show the disjoint set of attributes $\mathcal{A}, \mathcal{A}'$ for the Zappos-ID/OOD datasets. For the Zappos-ID dataset we use the set of attributes in $\mathcal{A}$ to *iid* sample train and test tasks $\mathcal{T}_a \sim \mathbb{P}_\mathcal{A}$. For the Zappos-OOD dataset the train tasks are still sampled using $\mathcal{A}$ *i.e.,* $\mathcal{T}_a \sim \mathbb{P}_\mathcal{A}$ but the test tasks are sampled using $\mathcal{A}'$ *i.e.,* $\mathcal{T}_{a'} \sim \mathbb{P}_{\mathcal{A}'}$.

| | | | | |
|---|---|---|---|---|
| $\mathcal{A}$ | Category.Boots | Category.Sandals | Closure.Ankle.Strap | Closure.Ankle.Wrap |
| | Closure.Buckle | Closure.Bungee | Closure.Button.Loop | Closure.Elastic.Gore |
| | Closure.Pull.on | Closure.Sling.Back | Closure.Snap | Closure.T.Strap |
| | Closure.Toggle | Closure.Zipper | Gender.Girls | Gender.Women |
| | HeelHeight.High.heel | HeelHeight.Short.heel | Material.Rubber | Material.Suede |
| | SubCategory.Ankle | SubCategory.Clogs.and.Mules | SubCategory.Flats | SubCategory.Heel |
| | SubCategory.Heels | SubCategory.Knee.High | SubCategory.Mid.Calf | SubCategory.Over.the.Knee |
| | ToeStyle.Almond | ToeStyle.Center Seam | ToeStyle.Closed Toe | ToeStyle.Open Toe |
| | ToeStyle.Peep Toe | ToeStyle.Pointed Toe | ToeStyle.Round Toe | ToeStyle.Snip Toe |
| $\mathcal{A}'$ | Category.Shoes | Category.Slippers | Closure.Hook.and.Loop | Closure.Lace.up |
| | Closure.Monk.Strap | Closure.Slip.On | Gender.Boys | Gender.Men |
| | Material.Corduroy | Material.Silk | Material.Wool | SubCategory.Boat.Shoes |
| | SubCategory.Crib.Shoes | SubCategory.Firstwalker | SubCategory.Loafers | SubCategory.Oxfords |
| | SubCategory.Prewalker | SubCategory.Slipper.Flats | SubCategory.Sneakers.and.Athletic.Shoes | ToeStyle.Algonquin |
| | ToeStyle.Apron Toe | ToeStyle.Bicycle Toe | ToeStyle.Bump Toe | ToeStyle.Capped Toe |
| | ToeStyle.Medallion | ToeStyle.Moc Toe | ToeStyle.Snub Toe | ToeStyle.Square Toe |
| | ToeStyle.Wide Toe Box | ToeStyle.Wingtip | | |

## D.2 Hyperparameter settings

For all the experiments performed in our paper, we have run grid search to tune both the meta-learning method-specific hyperparameters and the optimization hyperparameters for each meta-learning method. For the existing OOD FSL benchmarks, we found that the best hyperparameters are often the same as what was reported in the original paper. Additionally, the absolute performance and performance orders we report on OOD FSL benchmarks match with other works after the hyperparameter tuning – indicating that we have been fair in representing each meta-learning method with its best hyperparameter setting. For our newly identified in-distribution benchmarks, we took care in tuning these hyperparameters for each method to ensure fairness of comparison.

In Table 4 we list the optimization and other algorithm-specific hyperparameters for each meta-learning method, dataset pair. For **optimization hyperparameters**, we describe

(a) the total number of epochs (each with 1000 iterations of gradient updates except for FEM-NIST, Zappos-ID, and Zappos-OOD where each epoch depends on number of training tasks);

(b) step (staircase) learning rate schedule (lr: $e_1(r_1) - e_2(r_2) - \ldots - e_n(r_n)$ where $r_i$ is the value of the learning rate and $e_i$ is the epoch number at which $r_i$ is first set);

(c) the number of tasks in a minibatch to compute one gradient update (task batch size or task BS);

(d) for all experiments we use SGD optimizer (Nesterov Momentum 0.9).

**Other meta-learning method specific hyperparameters**: The scale-factor refers to a constant factor that is multiplied to the logits for each class, before passing them through softmax. In some cases these are fixed through training, and in others they are learnable. For other method-specific hyperparameters that we borrow directly from the original paper, we provide the references.

i) During meta-training, we perform the same **data augmentation** used in Chen et al. [9] for *mini*-M and used in Lee et al. [25] for *cifar*-M, FC-M, *tiered*-M. For FEMNIST, Zappos-ID and Zappos-OOD we do not perform any data augmentations.

ii) We use a **weight decay** of $5e-4$ for all datasets except for FEMNIST for which a weight decay of $1e-2$ is used to prevent overfitting.

iii) We use the Resnet-12 **backbone** for all our experiments except for FEMNIST which is made up of the relatively easier tasks of digit classification. For FEMNIST, we use a four layer Conv-64 model backbone.

**Computational Resources.** For all experiments we use (at most) four NVIDIA GEFORCE GTX 1080Ti GPU cards. A single run of PN, RR, SVM, and FOMAML on the *mini*-M dataset takes $\approx 12$ hrs, 48 hrs, 48 hrs, and 72 hrs respectively. For experiments on Zappos and FEMNIST, except for FOMAML which takes about 24 hrs, all the other experiments take no more than 5 hrs to complete training.

Table 4: Hyperparameter details for different algorithms and datasets in Sections 4, 5.

| Alg / Dataset | Optimization hyperparameters | Other hyperparameters |
|---|---|---|
| PN/ (*mini*-M, *cifar*-M, *tiered*-M) | 60 Epochs
lr: 0(0.1)-20(6e-3)-40(1.2e-3)-50(2.4e-4)
task BS: 4(5-way), 1(>5-way) | scale-factor 10
euclidean metric [39] |
| PN/ FC-M | 15 Epochs
lr: 0(0.1)-5(6e-3)
task BS: 4(5-way), 1(>5-way) | scale-factor 10
euclidean metric [39] |
| (RR, SVM)/ (*mini*-M, *cifar*-M, *tiered*-M) | 60 Epochs
lr: 0(0.1)-20(6e-3)-40(1.2e-3)-50(2.4e-4)
task BS: 8 (always 5 way) | learnable scale
other as in Lee et al. [25] |
| (RR, SVM)/ FC-M | 30 Epochs
lr: 0(0.1)-20(6e-3)
task BS: 8 (always 5 way) | scale-factor 7
other as in Lee et al. [25] |
| FOMAML/ (*cifar*-M, FC-M) | 60 Epochs
lr: 0(0.01)-20(6e-3)-40(1.2e-3)
task BS: 4 (always 5 way) | scale-factor 1
inner loop step size: 0.01
inner loop steps: 5 (train), 20 (test)
other as in Finn and Levine [14] |
| FOMAML/ (*mini*-M, *tiered*-M) | 70 Epochs
lr: 0(1e-2)-35(1e-3)-65(1e-4)
task BS: 4 (always 5 way) | scale-factor 1
inner loop step size: 0.01
inner loop steps: 5 (train), 20 (test)
other as in Finn and Levine [14] |
| (PN, RR, SVM, FOMAML)/ (Zappos-ID, Zappos-OOD) | 60 Epochs
lr: 0(0.1)-30(6e-3)
task BS: 4 | PN: scale-factor 10, euclidean metric [39]
RR: learnable scale
SVM: learnable scale
FOMAML: inner loop steps: 5 (train), 20 (test), other as in Finn and Levine [14] |
| (PN, RR, SVM, FO-MAML)/ FEMNIST | 100 Epochs
lr: 0(1e-3)-50(1e-4)
task BS: 5 | PN: scale-factor 10, euclidean metric [39]
RR: learnable scale
SVM: learnable scale
FOMAML: inner loop steps: 5 (train), 20 (test), other as in Finn and Levine [14] |
| (SB, MB)/ (*mini*-M, *cifar*-M, *tiered*-M, FC-M) | 100 Epochs
lr: 0(0.1)-90(1e-2)
batch size: 128 | SB, MB: We project features in to unit norm during meta-train and use euclidean metric with scale-factor 10 in meta-test [39]
MB only: we further finetune (lr=1e-3) on the meta-learning objective for additional 30 epochs. |
| IC/ (*cifar*-M, *mini*-M) | 100 Epochs
lr: 0(0.1)-90(1e-2)
batch size: 128 | Figure 2(b) (*cifar*-M): For supervised learning image classification (IC) baseline we collect all the images belonging to the base classes in *cifar*-M and randomly split them into 80%(train)-10%(val)-10%(test). The val and test rankings used for computing the Kendall coefficient ($\rho$) are obtained using the val and test splits.
Figure 3(b) (*mini*-M): The supervised learning IC baseline is trained on 600 train images from each base class in *mini*-M. After training is completed, we identify two IC models that differ by $\epsilon = 0.5\%$ in terms of their generalization performance over a test set made up of all the unused examples ($\sim 700$) from each base class. To test the frequency of conclusion flips, for each of the 100 comparison runs, a random test subset is sampled from this test set with 100 examples sampled from each class. The chosen IC model pair is then evaluated over this test subset and their performance difference is recorded for this comparison run. |

# E    Additional Results on Evaluating ID Performance

In this section, we first present additional results on the choice of a different meta-learning method or the usage of a different $(S, Q)$ sampling strategy (FIX-ML) for the same meta-learning method can lead to improvements in OOD performance at the cost of ID performance. Then, we show how reducing the number of training tasks is unlikely to change the performance order of meta-learning methods in the ID benchmark Zappos-ID — even though by reducing the number of training tasks, the number of unseen attribute pairs at test time increases. Finally, we present additional results how the degree of train/test task distribution mismatch can impact the performance order of meta-learning methods.

## E.1    Additional results on ID/OOD Performance Tradeoffs

In the main text (Section 4, Figure 1), we have compared the BaseGen and NovelGen performance of the best validation snapshots from different meta-learning methods (PN, SVM, RR, FOMAML) and supervised trained baselines (SB, MB) on the *mini*-M and *tiered*-M datasets. Here, we show how the performance order of the same set of methods *can also flip* in the ID (BaseGen) and OOD (NovelGen) settings **on two additional datasets**: *cifar*-M and FC-M (Figure 4(a),(b)). Additionally, in the main text, we have shown how switching the choice of the $(S, Q)$ sampling strategy to one with fixed support sets (FIX-ML Setlur et al. [38]) can also lead to improvements in the OOD performance at the cost of ID performance on the *cifar*-M dataset. In Figure 4(c),(d) we provide further evidence of this performance tradeoff **on two more datasets** *mini*-M and FC-M.

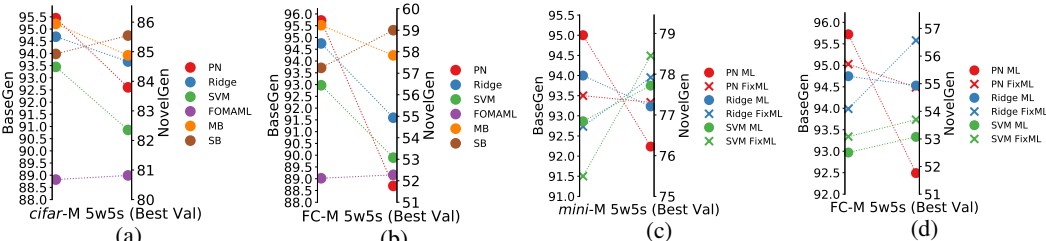

Figure 4: BaseGen and NovelGen performance tradeoff (for best validation snapshots): over the choice of a set of four meta-learning and two supervised pre-training methods on *cifar*-M (a) and FC-M (b); over the use of FIX-ML $(S, Q)$ generation strategy or not (ML) with SVM, RR and PN on *mini*-M in (c) and FC-M in (d).

## E.2    In-distribution Performance with Reduced Number of Training Tasks

Now we analyze the impact of the number of training tasks on the performance order of meta-learning methods in ID scenario. For the 2w5s results on the Zappos-ID dataset in Table 1 we used 1000 (or 50) training tasks and 25000 test tasks. Here, we also consider using 250 training tasks (while still using the same number of test tasks). As we have discussed in Section 4, it is possible to encounter unseen classes (attribute pairs in the case of Zappos) at test time even while evaluating a meta-learning method in the ID scenario. When we reduce the number of training tasks, we observe more unseen attribute pairs at test time. For 250 train tasks we observe 153 and for 50 train tasks we observe 269 unseen attribute pairs at test time.

Table 5: We analyze the 2w5s performance order of PN, SVM, RR and FOMAML on the Zappos-ID dataset with reduced number (50, 250) of train tasks and compare the performance order (ranking in parentheses) observed with the larger set of 1000 train tasks.

| Methods / # Train tasks | 50 | 250 | 1000 |
|---|---|---|---|
| PN | [1]$77.67 \pm 0.17\%$ | [1]$81.67 \pm 0.16\%$ | [1]$86.58 \pm 0.15\%$ |
| Ridge | [2]$74.75 \pm 0.16\%$ | [2]$80.84 \pm 0.15\%$ | [2]$85.56 \pm 0.16\%$ |
| SVM | [3]$74.06 \pm 0.17\%$ | [3]$80.15 \pm 0.17\%$ | [3]$85.12 \pm 0.16\%$ |
| FO-MAML | [4]$69.85 \pm 0.18\%$ | [4]$73.20 \pm 0.16\%$ | [4]$80.14 \pm 0.15\%$ |

In Table 5 we can see that even with reduced training tasks and more unseen attribute pairs at test time the performance order of PN, RR, SVM and FOMAML is retained. Note that the same order is observed on the other ID benchmark FEMNIST. This also matches the BaseGen performance order obtained after doing ID evaluations on modified FSL benchmarks. This result further confirms that the performance evaluations done on ID datasets are much more consistent than OOD datasets.

### E.3 Degree of Mismatch between Train ($\mathbb{P}_{\mathcal{C}_B}$) and Test ($\mathbb{P}_{\mathcal{C}_N}$) Distributions

In Section 4 we notice that the performance order of meta-learning methods in the ID scenario (Zappos-ID) is quite different from that of the OOD FSL benchmarks (*e.g., mini*). We believe this is mainly because of the OOD nature of FSL benchmarks, which we also proof formally in Appendix B. Moreover, since the type/degree of the mismatch between training and test distributions can vary for different FSL benchmarks, the performance order of popular methods is not as consistent as we would like them to be on these OOD benchmarks (see Section 5.2).

**Range of test task distributions.** To further analyze the impact of the degree of distribution mismatch on the performance order, for each FSL benchmark we construct **a range of new task distributions that are increasingly dissimilar to the train task distribution** $\mathbb{P}_{\mathcal{C}_B}$ and **similar to the test task distribution** $\mathbb{P}_{\mathcal{C}_N}$. We denote this set of task distributions as $\{\mathbb{P}_{\text{BN}}(\lambda), \lambda \in [0,1]\}$. When sampling an $n$-way task from $\mathbb{P}_{\text{BN}}(\lambda)$, instead of performing the first step of sampling outlined in Table 2, we sample the class tuple $\boldsymbol{c} = (c_1, \ldots, c_n) \in [(\mathcal{C}_B \cup \mathcal{C}_N)^n]$ where $c_i$'s are sequentially sampled non-repeatedly from $\mathcal{C}_N$ uniformly with probability $\lambda$ and from $\mathcal{C}_B$ with probability $1 - \lambda$. The second step of sampling $(S, Q)$ from the task distribution $\mathcal{T}_{\boldsymbol{c}}$ is done in the usual way. It is clear that $\mathbb{P}_{\text{BN}}(0) = \mathbb{P}_{\mathcal{C}_B}$ and $\mathbb{P}_{\text{BN}}(1) = \mathbb{P}_{\mathcal{C}_N}$. In addition, when $\lambda = |\mathcal{C}_N|/(|\mathcal{C}_N| + |\mathcal{C}_B|)$, $\mathbb{P}_{\text{BN}}(\lambda) = \mathbb{P}_{\mathcal{C}_B \cup \mathcal{C}_N}$ which describes a continual learning setting where the evaluation task is made up of classes uniformly sampled from the union of base and novel classes. Notice that the task distribution $\mathbb{P}_{\text{BN}}(\lambda)$ is not the same as the mixture distribution $\lambda \cdot \mathbb{P}_{\mathcal{C}_N} + (1 - \lambda) \cdot \mathbb{P}_{\mathcal{C}_B}$ because a single task from $\mathbb{P}_{\text{BN}}(\lambda)$ can have both classes from $\mathcal{C}_N$ and $\mathcal{C}_B$. With this set of new task distributions defined, we evaluate our learned algorithm snapshot for each meta-learning method not only over the training ($\mathbb{P}_{\mathcal{C}_B}$) and the test ($\mathbb{P}_{\mathcal{C}_N}$) task distribution but also over distributions from this interpolation set. We plot the performances in Figure 5.

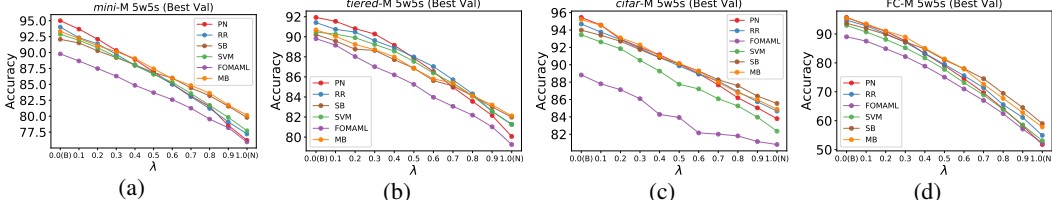

Figure 5: Comparison of PN, SVM, RR, FOMAML, SB and MB's performance (best validation snapshots) on the set of distributions $\{\mathbb{P}_{\text{BN}}(\lambda)\}$ for (a) *mini*-M, (b) *tiered*-M, (c) *cifar*-M, and (d) FC-M datasets.

**Performance order depends on degree of mismatch.** In Figure 5 we first notice that the performance drops in a monotonically non-increasing way as $\lambda$ increases for each dataset/method combination. More importantly, we note that the performance order of the methods depends on the degree of mismatch. If the test task distribution is very similar to the train task distribution, then the performance order is mostly retained as the ID performance order (*e.g.,* for $\lambda \leq 0.2$ in Figure 5(a)). Also in most cases the lines don't cross each other more than once, which indicates that if the degree of mismatch crosses a certain threshold then it is unlikely for the performance order to switch again for a given pair of methods. However, without seeing any test tasks during meta-training, it is difficult to know the degree of training and test task distribution mismatch, which makes it more difficult to predict when a given method will start performing better/worse. For example, the performance order changes at a lower value of $\lambda$ for *mini* and *cifar*, as compared to *tiered*.

## F Additional Discussion and Results on OOD Evaluation

### F.1 Simplified Example of Snapshot Selection and Hyperparameter Selection

To further explain our existing definition of hyperparameter and snapshot selection in Section 5.1, we accompany our original definitions with a concrete simplified scenario: suppose we want to train

Prototypical Network (PN) with two hyperparameter configs: training with learning rate 1e-3 vs. with learning rate 1e-4 (with all other hyperparameters the same).

**[Snapshot Selection]** Training under each hyperparameter config would generate a sequence of algorithm snapshots (most often with one snapshot saved after each training epoch). The term snapshot selection refers to the procedure of choosing one snapshot from each hyperparameter config: one from the 1e-3 learning-rate optimized PN algorithm snapshot trajectory, and one from the 1e-4 learning-rate optimized PN algorithm snapshot trajectory. There can be multiple strategies for snapshot selection, for example, picking **1)** the last snapshot at the end of training; **2)** the snapshot that has the lowest training loss; **3)** the snapshot that has the best BaseGen performance; or **4)** the snapshot that has the best ValGen performance.

**[Hyperparameter Selection]** Once an algorithm snapshot is chosen for each considered hyperparameter config, we need to decide which hyperparameter config's (lr 1e-3 or lr 1e-4) selected snapshot to choose to be evaluated on the test task distribution with its performance recorded as the meta-learning method PN's performance. The procedure of deciding which hyperparameter config's snapshot to choose is called the hyperparameter selection problem. Similar to snapshot selection, there can be multiple strategies for hyperparameter selection: e.g., choosing the hyperparameter config whose selected snapshot has the best BaseGen or the best ValGen performance.

## F.2 Distinction between Snapshot Selection and Early Stopping

*Early stopping is not the same as what we mean by snapshot selection in our paper.* In standard supervised learning, *early stopping* involves keeping track of the performance over an *iid* validation dataset and stopping training after the validation performance starts to consistently deteriorate. It mainly serves as an approach to avoid overfitting and to save unnecessary computations if one believes further training would never improve the test performance. In terms of deciding which snapshot to choose, early stopping is often equivalent to the strategy of selecting the snapshot that has the highest validation accuracy. In contrast, in our paper, the term *snapshot selection* refers to the **general problem** of deciding which snapshot to select for a given training run, **as opposed to a specific choice of selection strategy**. As we have explained in Section 5.1.1, the strategy of picking the snapshot with the best meta-validation performance (early stopping) might not be the best strategy in the OOD scenario. There exist other snapshot selection strategies (different from early stopping), such as picking the snapshot with the lowest training loss or the best BaseGen performance.

## F.3 Differences in NovelGen Performance When Using Best BaseGen vs. ValGen for Snapshot Selection

In Section 5.1 we compare different model selection strategies in the out-of-distribution scenario where we show that in some cases BaseGen performance can track NovelGen (Figure 2(c)) while in other cases the ValGen performance may be better correlated with NovelGen (Figure 2(d)). In Table 6 we present NovelGen (test) performance results on two different datasets when the snapshot is chosen using the best BaseGen snapshot (row 1) vs the best ValGen snapshot (row 2), and compare them against the best possible NovelGen across all snapshots (row 3). Here we notice that there can be a **significant difference in the selected snapshot's NovelGen performance if we use one selection strategy instead of another** and there isn't a single snapshot selection strategy that yields the best results for multiple OOD scenarios.

Table 6: NovelGen performance for i) PN trained on *tiered*-M; and ii) SVM trained on *cifar*-M evaluated using the snapshot chosen with best BaseGen/ValGen/NovelGen performance.

| Snapshot selection strategy | PN on *tiered*-M (Figure 2(c)) | SVM on *cifar*-M (Figure 2(d)) |
|---|---|---|
| Best BaseGen | 80.45% | 80.49% |
| Best ValGen | 80.05% | 82.65% |
| Best NovelGen | 80.45% | 82.79% |

### F.4 What to Consider While Designing Model Selection and Comparison Strategies for Both ID and OOD Performance?

In Section 5.1 we discuss in detail the implications of different model selection strategies on the final NovelGen performance of the selected snapshot specific to current OOD FSL benchmarks. In more generic settings, it is likely that one may wish to design model selection methods when one cares about both ID and OOD performance. In such a case we believe that the optimal model selection and comparison strategy would depend on: i) the final comparison metric; ii) the type of task access for model selection.

i) **Method comparison metric**: We first need to define a metric for the final evaluation of a meta-learning method's selected algorithm snapshot. One way is to individually evaluate ID and OOD performance and record it as a 2-tuple. In this case, a meta-learning method is said to outperform another only if its selected snapshot is better in each component of the 2-tuple. In this case, it is very possible that there does not exist a meta-learning method that clearly outperforms all others in this metric as we have seen from Figure 1(a)(b). Another way is to evaluate the performance on a mixture task distribution, with the training and OOD test distributions weighted by fixed probability weights (as we do in Appendix E.3). As we have seen in Appendix E.3 (Figure 5), different probability weighting of the two distributions can result in different conclusions of which meta-learning method works the best.

ii) **Task access during model selection**: We also need to specify what type of task samples are available during model selection. While it is reasonable to assume there are additional fresh *iid* samples from the training distribution (e.g. by holdout training set), it depends on the specific application to know whether there exist task samples from the OOD task distribution. Different scenarios may arise based on this: i) In federated learning applications, one might be allowed to evaluate meta-learned algorithm snapshots over a small sample of users (tasks) from the OOD user population before deployment; or ii) In case one does not have such OOD task samples, a proxy distribution (e.g. validation task distribution in FSL benchmarks) may still be available and *iid* task samples drawn from it could be used for model selection. However, its utility would depend on how similar it is to the actual OOD distribution and we have seen examples in Figure 2(a) that using samples from such proxy task distributions might also be suboptimal.

We hope that our work advocates for more discussion and development of model comparison metrics and selection strategies while taking into these considerations described above.

### F.5 Hyperparameter Selection Strategies

In Section 5.1 we discuss the distinction between snapshot selection and hyperparameter selection and how the former is called as a subroutine while determining the best snapshot to represent a specific hyperparameter configuration. As snapshot selection is a ubiquitous problem we focus on analyzing it and exploring alternative strategies (in the context of meta-learning) in the main paper. Motivated by the work of Gulrajani and Lopez-Paz [18], we now discuss some ways of performing hyperparameter selection specifically for settings similar to the current FSL benchmarks where the few-shot classification tasks are determined by a class tuple.

Because the current FSL benchmarks have set aside a specific validation set of classes, it would appear that using tasks generated by these classes is the only option for hyperparameter selection. However, as mentioned in [18], there are also other alternatives.

**Cross validated hyperparameter selection**. Instead of using a single set of validation classes, one could rely on cross validation. Here, for a given hyperparameter configuration, one can train multiple times, where each training run is done on tasks generated by a different subset of the training (base) classes (or the union of base and val classes) and the proxy performance is calculated on tasks generated by the remaining classes not used in training. Finally, the performance is averaged over all runs for the given hyperparameter configuration and the hyperparameter with the highest performance would be chosen.

However, this approach would still require performing snapshot selection for each run, thus it remains unclear how to best perform snapshot selection in this case. Despite this, cross-validated hyperparameter evaluation could potentially be more reliable than using a single set of validation classes for hyperparameter selection (as done on the current FSL benchmarks) but it would also be

more computationally expensive and we leave further investigation of this hyperparameter selection approach as future work.

**Allowing restricted oracle evaluation over the test task distribution.** Instead of completely forbidding access to the test task distribution, one can allow a limited number of test task distribution performance evaluations for a given meta-learning method. In this case it becomes the responsibility of designer of the meta-learning method to decide how to best distribute this fixed number of evaluations wisely over different hyperparameter and snapshot choices.

## F.6 Variance of performance $A(\mathcal{C}_N)$ over randomly sampled $\mathcal{C}_N$

In Section 5.2 Inconsistency example 1, we have shown how limited number of novel classes in the evaluation can lead to a high chance of conclusion flips. This high degree of unreliability stems from the variance of the performance estimator $A(\mathcal{C}_N)$ which only uses a subset of the larger underlying class set $\mathcal{C}_L$. In Figure 6 we plot the histogram of the random variable $A(\mathcal{C}_N)$ randomized over the choice of novel classes $\mathcal{C}_N$. We notice that the performance standard deviation is $2.49\%$ on *mini* and $3.1\%$ on *tiered*. The standard deviation on *tiered* is higher since the number of underlying classes $|\mathcal{C}_L|$ is larger in *tiered* ($= 552$). When the variance of the novel accuracy is as high as what we have observed here, it becomes very hard to clearly determine the better meta-learning method. To alleviate this, we provide some actionable suggestions like choosing benchmarks with more base classes during training and more novel classes during evaluation (see Section 5.2).

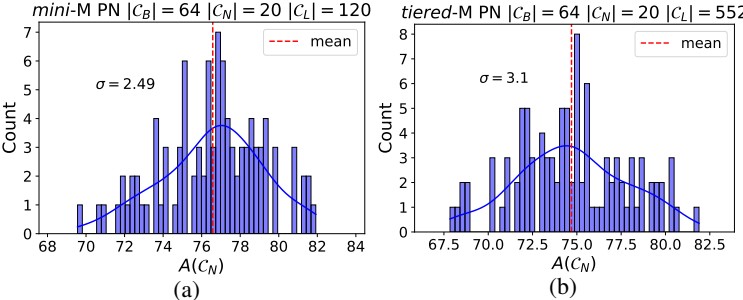

Figure 6: Histogram plots (over 100 runs) of the OOD accuracy $A(C_N)$ for (a) a PN trained on *mini* and evaluated on random 20 out of 120 novel classes; and (b) a PN trained on 64 *tiered* base-classes (see Section 5.2) and evaluated on random 20 out of 552 novel classes.