# OpenReview forum: "Two Sides of Meta-Learning Evaluation: In vs. Out of Distribution"
_NeurIPS.cc/2021/Conference — NeurIPS 2021 Poster_

### Official Review · Reviewer_mSBf · 2021-07-12

**Rating:** 5
**Confidence:** 3

**Summary:**

This paper claims that few shot learning / meta learning benchmarks are testing a form of OOD generalization and are not testing generalization to IID sampled tasks from a task distribution. The main observation is that most benchmarks train and test on tasks with disjoint sets of classes, which is improbable if the label spaces for each task are sampled IID. The authors convert benchmarks into a "ID" setting by re-sampling the dataset and show that there is a tradeoff in performance between the settings. In particular, they find that methods that perform well on the benchmark (which is testing OOD generalization) may perform worse in the ID setting, and they note that there are some challenges when doing model selection in the OOD setup.

**Limitations And Societal Impact:**

The paper does not address the limitations / social impacts of the work.


**Main Review:**

Quality/Originality

The discussion of ID/OOD evaluation in FSL datasets is insightful, and the authors took care to hold out examples in each class for ID evaluation (now that classes can appear in both train and test). The same model backbone is used throughout for comparison between methods. However, main concerns are that the comparisons don't seem to be run on all the datasets considered, and the picture would be more complete with these, and the conclusions given throughout the paper are somewhat inconsistent. Clarity on these would improve my score.

- The observations could be bolstered by running more datasets using the method in Figure 1a,b, where each dataset is converted into ID/OOD evaluations and the numbers are compared. For example, in Figure 1d, there is still inconsistency but only for 2 of 5 methods. What happens if you convert Zappos and FEMNIST to OOD evaluations as well? The tables comparing the rankings across datasets are valuable but less of a controlled setting.
- Similarly, in line 427, the paper claims a contrast between the results in Figure3c and Table 1. However, we only have one example of BaseGen comparisons between datasets and within datasets in Table 1, as well as one comparison between BaseGen and OOD in Figure 3c, when it seems that we should be able to make these comparisons between all the datasets, given that they can all be converted into ID/OOD versions.
- The message between Section 4 and 5 are somewhat contradictory at times. For example, Section 4 describes an ID/OOD conflict, but Section 5 says that using BaseGen(ID) as a model selection measure for NovelGen(OOD) should be considered.
- The paper says that current meta learning methods are worse in the ID version of the dataset than the OOD (normal) version. Can the authors explain why methods developed using theoretical/conceptual insights from the ID setting are worse in the ID setting than OOD? Even though there could be an effect from tuning the models, at a fundamental level shouldn't ID be easier and match the statistical nature of the methods better? Some discussion of this in the paper may be useful.
- As stated in line 134, another possible explanation for disjoint classes between train and test tasks is that there is a much larger ambient set of classes. For consistency, suppose that the benchmarks just don't have enough training tasks for this ambient space of tasks. In this case, the ID theory/setup holds but the benchmarks are not totally fair, which may lead to inconsistent results. This explanation doesn't seem to be rejected by this paper? The remedies may be different in this case.
- ID evaluation using BaseGen (accuracy on held-out examples from base classes that were sampled from for training) for meta-learning datasets with a small number of classes (such as mini) may not test generalization to any new classes, so this could exaggerate the difference between ID and OOD for these datasets.
- The observation in L256 that higher learning rate is better for BaseGen while lower learning rates are better for OOD generalization (NovelGen) are counterintuitive since higher learning rate is like regularization. Is there any explanation for this?



Clarity

The explanation of the why benchmarks seem more like OOD evaluation is clear. Sections 4/5 were less clear in terms of organization - while Section 4 seems to be about ID evaluation, the main points are about ID vs OOD, which muddles the distinction between Sections 4 and 5. They both make statements about inconsistency and ID vs OOD, but sometimes (as stated above) the messages seem somewhat at odds, and muddles the picture.

- Some common quantities seem to be renamed - for example, early stopping is renamed to snapshot selection. I think using the common terminology could make things clearer.
- The definition of BaseGen/ValGen/NovelGen can be made clearer - at first glance, BaseGen sounds like training accuracy, but actually it's held out examples from the training classes.


Significance

Typically, few-shot learning / meta-learning methods are developed with the assumption that tasks are sampled IID from some task distribution. This paper shows that this common assumption is not satisfied in the empirical setup of few-shot benchmarks, which may have implications on empirical evaluation of the methods. A potential outcome is to evaluate meta-learning/few-shot methods in both the original (OOD) setting and a modified ID setting. This would encourage methods that would work in both of these evaluation settings.

**Time Spent Reviewing:**

4

---

> ### Author Response · Authors · 2021-08-10
> **Response to Reviewer mSBf**
>
> **[Bolstering observations on ID vs OOD ranking inconsistency]**
>
> - **[More observations beyond Fig1(a),(b)]** In terms of results on more datasets, we have provided the ID vs OOD performance flips results not only on the mini and tiered shown in Figure 1(a),(b) but also on CIFAR-FS and FC-100 in Appendix E.1 Figure 4(a)(b). We believe observing consistent ID vs OOD performance flips over these 4 popular benchmarks should already substantiate our claim.
>
> - **[Figure 1(d)]** is not intended for comparing different meta-learning methods. Instead we show that a different task sampling strategy FixML, when applied to each of the three meta-learning methods, can only improve OOD performance at the expense of ID performance (L259-265). In Figure 1(d), the comparison should be drawn between lines of the same color but not the same method. We also have additional results on this sampling strategy with more datasets in Appendix E.1 Figure 4(c)(d).
>
> - **[FEMNIST and Zappos-OOD]** Because there is no controllable way to create out-of-distribution writers, FEMNIST cannot be easily made into an OOD benchmark. We’ve provided Zappos-OOD (2w5s, 2w10s) results in Figure 3(c) where the performance ranking is consistent with Zappos-ID in Table 1. However, this does not invalidate our claim that there exists multiple scenarios where ID/OOD performance orders can flip, because we never claim the performance ranking would always flip. Instead, we believe that the performance ranking inconsistencies observed over the 4 popular FSL benchmarks sufficiently highlight the ID/OOD evaluations’ distinctions.
>
>
>
> **[What do we mean in Line 427]** We think you might have misunderstood our arguments in L427. We claim that over all six ID benchmarks we’ve experimented with (FEMNIST, Zappos, modified mini-M, tiered-M, cifar-M, FC-M), the ranking order of the four meta-learning methods are always consistent. However, for the two different types of OOD datasets in Figure 3(c), the rankings are flipped. This shows that in different OOD scenarios, the best meta-learning method can be different, which is unlike all the different ID scenarios we’ve experimented with where PN is always the best method.
>
>
> **[Are Sec 4,5 contradictory at all?]** In Sec 4, we claim and show that the performance order of *independently* trained **different** meta-learning methods in ID and OOD can flip. This does not necessarily imply that BaseGen and NovelGen performance can’t be correlated over the *same set* of algorithm snapshots along the optimization trajectory (hence dependent set of snapshots) of a **single** meta-learning method. As we show in Fig 3(c), for snapshots along the optimization trajectory of PN trained on tieredImagenet, NovelGen and BaseGen are still correlated. Thus snapshot selection according to best BaseGen performance (Section 5) is still a valid suggestion and is not contradictory to our findings in Section 4.
>
>
> **[ID worse than OOD?]** We never claim that ID performance of a meta-learning method is worse than its OOD performance. Instead, we show the opposite in Fig 1, where the ID performance of each method (value on the left) is always better than OOD performance (the value on the right connected by the same line) -- notice that the left and right axes in Fig 1 have different ranges of values. We believe you might have misunderstood our original claim in L36-37, L239-241 as referring to the comparison between the ID performance value and the OOD performance value of the same meta-learning method, while actually the comparison we made is between the ID performance ranking and OOD performance ranking of the same **set** of meta-learning methods.
>
>
> **[ID explanation cannot hold even with fewer tasks and larger ambient set of classes]** If you were to believe that there is a much larger set of ambient classes and the training tasks are sampled *iid* from the induced task distribution, then the classes in different *iid* task samples would almost never overlap. As a result, the number of classes seen during training would scale roughly with the number of tasks sampled with high probability (Appendix B, L746-748). Because the common practice is to use more than $10^4$ tasks for training, with high probability under the ID assumption we would have seen a lot more base classes than 64 (mini) or 351 (tiered) during training, thus creating a contradiction. Our formal proof (L139-142) and Appendix B.4 already reject your proposed ID explanation.
>
>
>
> **[ID evaluation using BaseGen]** We agree that since mini has very few (64) base classes, the 60,000 tasks we sample (uniformly) during training from the base classes would cover all 64 classes. However, it would not cover all possible tasks (class combinations) in the training task distribution since there are $64 \choose 5$ $\approx 7.6 \times 10^6$ distinct 5-way tasks (unordered class tuples). Thus, BaseGen evaluation would still test generalization to unseen tasks with different class combinations. Beyond ID evaluation using BaseGen, for the Zappos-ID dataset where we only have 1000 meta-train tasks, we have shown (L194-200) that the learned algorithm would still encounter unseen attributes at meta-test time. Even in this case, we see a significant flip in performance order when compared to OOD FSL benchmark (Table 1 Zappos 2w5s).
>
>
>
>
> **[Why does higher learning rate work better for BaseGen but lower learning rate work better for NovelGen?]** The common wisdom that higher learning rate works as regularization and thus would prevent overfitting technically only applies to in-distribution generalization (Cheng et. al. [see below]) but not necessarily to out-of-distribution. Here we do see that using higher learning rate ensures better in-distribution generalization which corroborates this intuition. However, when we switch to the OOD evaluation, it is possible that a more complex solution (obtained using a lower learning rate) could fit the unseen test task distribution better than a simpler solution (obtained using a higher learning rate). Thus training with a lower learning rate and overfitting the training tasks might prove to be sometimes beneficial for OOD generalization.
>
> Cheng, X., Yin, D., Bartlett, P., \& Jordan, M. (2020, November). Stochastic gradient and langevin processes. In International Conference on Machine Learning (pp. 1810-1819). PMLR.
>
>
>
> **[Clarity - Why discuss ID vs OOD in Section 4?]** As we introduce realistic FSL examples that require ID generalization in Section 4, we believe it is natural to compare the existing meta-learning methods over these new benchmarks. From there we discovered the phenomenon of ID/OOD performance ranking inconsistency which then motivated our additional analyses in modifying the OOD FSL benchmarks to also enable ID evaluation to further verify this inconsistency. We believe all these analyses fit well in Section 4 because it gives further evidence that ID evaluation is a paradigm very distinct from OOD evaluation.
>
>
> **[Clarity - early stopping vs snapshot selection]** Early stopping is not the same as what we mean by snapshot selection. In standard supervised learning, **early stopping** involves keeping track of the performance over an *iid* validation dataset and stopping training after the validation performance starts to deteriorate. It mainly serves as an approach to avoid overfitting and to save unnecessary computations if one believes further training would never improve the test performance.  In terms of deciding which snapshot to choose, early stopping is often equivalent to the strategy of selecting the snapshot that has the highest validation accuracy. In contrast, in our paper, the term **snapshot selection** refers to the general problem of deciding which snapshot to select for a given training run, as opposed to directly specifying a specific strategy for use. Besides, as we mentioned in L316-326, the strategy of picking the snapshot with the best meta-validation performance (early stopping) might not be the best strategy in the OOD scenario. And there exist other snapshot selection strategies (different from early stopping), such as picking the snapshot with the lowest training loss or the best BaseGen performance.
>
>
>
> **[Definition of Base/Val/NovelGen]** Definitions for these are stated in L234-236 using terminology defined in L155-156. We will try to make this more clear.

---

> > ### Comment · Reviewer_mSBf · 2021-08-13
> > **response**
> >
> > Thanks to the authors for the clarifications. I think the main ID vs OOD evaluation point is valuable, and there's probably inconsistency going on.
> >
> > Reading the other comments, it seems that the main point from many reviewers was that the results should be supported by more datasets to make sure that the trends are solid. The authors don't agree, but I think most of the reviewers feel the same on this. About the effect sizes, while it is true that the effect size is generally pretty small (and more datasets could improve confidence on this), I think the point made in the common response is good, that with fewer training tasks the gap is larger and more clear.

---

> > > ### Author Response · Authors · 2021-08-21
> > > **Additional Response to Reviewer mSBf**
> > >
> > > Thank you for your response! We are not sure if we fully understand your concern regarding “*results should be supported by more datasets*”. It would be helpful if you could clarify which one of our claims should be backed up by more datasets. If you are asking the same question as Reviewer rpPm to verify whether different ID benchmarks rank meta-learning methods more consistently than different OOD benchmarks, please refer to our reply to Reviewer rpPm [More datasets to show ID ranks methods more consistently than OOD].
> > >
> > > To recap the datasets we used in our paper:
> > > - For in-distribution evaluation, we use six different benchmarks; two new realistic benchmarks (Zappos-ID and FEMNIST) that we propose to specifically target ID few-shot learning evaluation and four benchmarks that are modified versions (to enable ID evaluation) of existing current OOD FSL benchmarks.
> > > - For out-of-distribution evaluation analyses, we used four of the most commonly used OOD FSL benchmarks which we believe are largely representative of the current OOD evaluation practices. (Many published methodology papers [3, 23, 28] have only used a subset of these four benchmarks).
> > >
> > > Beyond this, we believe your summary that “*most of the reviewers*” think that the results should be supported by more datasets is not fully accurate: Reviewer qFSe described that our paper has “*strong supporting experiments*”, while reviewer Qp31 says that our paper “*contributes thorough empirical analyses*”.

---

### Official Review · Reviewer_Qp31 · 2021-07-15

**Rating:** 5
**Confidence:** 4

**Summary:**

This paper proposes to distinguish two types of evaluation settings for meta-learning algorithms: in-distribution (ID) and out-of-distribution (OOD). They claim that using few-shot learning benchmarks as a test bed for meta-learning algorithms leads to developing models for and studying only OOD, whereas other real-world settings require ID generalization, and the developed models may not work well there out-of-the-box. They investigate the trade-offs between ID and OOD performance of different models by creating a set of ID tasks and demonstrating different rankings between models on ID versus OOD. They also discuss other challenging facets of the evaluation protocol for meta-learning algorithms, including how model selection is performed and the reliability of conclusions drawn when an insufficient number of classes are used to create test tasks.


**Limitations And Societal Impact:**

The authors have not addressed these aspects. Even if not directly applicable to this paper specifically, I would encourage them to think about the potential negative impacts of meta-learning and few-shot learning works more broadly.

**Main Review:**

High-level review
-----------------------
The paper is well-written and easy to understand for the most part. The discussion around ID and OOD tasks is interesting and alerting the community that models developed for one might not work optimally for the other is a useful insight and a nice contribution. The paper also contributes thorough empirical analyses on various facets of meta-learning evaluation that are interesting. However, I have a number of concerns about the paper, summarized here (with more details in the section below). First, despite performance differences and differences in ranks between models in ID and OOD pointed out by the authors, these differences are small. I think a more thorough evaluation is needed to establish the performance differences of various models in these two settings. Along the same lines, the proposed ID setting is very similar to some related work (“low-shot” or “incremental” learning) which is cited but without comparing and consolidating insights with the findings of that community (e.g. about the nature of methods that work well in one setting versus the other) which would have been very useful. Finally, some topics investigated seem inconclusive and I’m unsure what the take-away should be (e.g. the model selection experiments) while others, albeit interesting, don’t seem particularly surprising, like the unreliability of performance comparisons between methods on benchmarks with small meta-test sets. Indeed recent benchmarks have moved towards larger-scale and more diverse test sets [2,3,4,8]. For these reasons at this stage I am recommending weak rejection despite the merits of the paper (more details and suggestions below).

Detailed comments
--------------------------
- [General] The narrative unnecessarily revolves around *meta-learning* evaluation / *meta-learning* methods specifically. The episodic evaluation protocols explored in this paper also can be used more generally for transfer learning approaches (and in fact some of the experiments include these models too). Perhaps using the term "few-shot learning evaluation" or "episodic evaluation" more appropriately captures the more general scope.

- [proof-by-contradiction] I don’t think the assumption about the lower bound on the number of training tasks (lines 135-136) is realistic. There could be enough tasks for training and yet still there could be a really large set of novel classes, so that the overall size of C_L is much bigger than the number of training classes, e.g. in Meta-Dataset when training on ImageNet only and evaluating on classes from a large number of other datasets.

- [ID vs OOD results] In line 212, the authors mention that the performance order of different models is consistent on ID benchmarks yet “almost completely opposite” on OOD ones. It should be noted though that in both settings the performance of these methods is really similar to each other (with the exception of FO-MAML which underperforms the rest consistently in both settings). This makes me unsure if the trend is significant, and I wonder, for instance, if different model selection techniques would eliminate this discrepancy. What was the criterion used for model selection in these experiments? Using a validation set of held-out *classes* is probably more appropriate for model selection for OOD evaluation, while using one of held-out images of *seen classes* is probably most appropriate for ID.

- [Ranking similarity analysis] I'm not sure that comparing the ranking is a good metric as it doesn't take into account the actual performance difference of the snapshots. For example, if one snapshot performs similarly to another, then their ordering in the ranked list does not really matter. Perhaps a more appropriate metric is a weighted average of the snapshots’ performance, with weights determined as a function of the ranks?

Modifying OOD FSL benchmarks for ID
-----------------------------------------------------
- Note that the modifications proposed for this are the same as those used in “low-shot” benchmarks [1]. Why not just re-use those benchmarks, which would also allow a broader comparison with few-shot learning models defined for that context?

- It should be noted that these ID benchmarks no longer reflect few-shot learning problems, since the classes used in test tasks are the same as those used for training. Therefore it might not be unexpected or surprising if a model developed for few-shot learning does not work well for a different problem. Though I agree it’s interesting to identify whether this is the case.

- A more compelling ID setting in my opinion (which actually can be thought of as few-shot learning still in some sense, as opposed to the proposed setting) is to first construct all possible N-way tasks (N-tuples of classes) and split those into train / validation / test tasks, following the “randomized tasks partition” scheme mentioned earlier. This modified variant can still be thought of as an ID "few-shot learning" setting: each given test task might contain classes that were seen at training, but those classes weren’t seen *together* (i.e. in the same task) during training.

“Challenges with Evaluation”
--------------------------------------
- The model selection analysis seems inconclusive. What is the take-away in terms of when to use one approach versus the other? As a general rule of thumb, does using a validation set of held-out classes work better as a model selection technique for OOD while a validation set of held-out examples of seen classes for ID?

- [Limited number of novel classes] I agree with the argument made about inconclusive results when a small number of test classes are considered. However, in addition to the *size*, I would argue that the *diversity* of the test set really matters to broadly evaluate different methods OOD. Even if we can reliably measure the performance difference on a specific (large enough) set of test classes, this doesn't necessarily speak to a model's broad generalization ability if the chosen set of test classes are similar to each other. Running an additional experiment to investigate this aspect would be really interesting too. This would present an argument in favour of using benchmarks with diverse sets of test classes like Meta-Dataset or VTAB [2,3,4,8].

- [Consistency across OOD FSL benchmarks] The authors point out the difference in relative performance of models between mini-OOD and Zappos-OOD from [5]. I would argue though that the setup in [5] reflects a different problem, not just a different benchmark. Specifically, they define a scenario where instead of having rigid class groupings that depend on the object classes (e.g. cats vs dogs), they define classes *flexibly* based on some hidden context (given by conjunctions of attributes). Therefore, images that belong to the same "class" in one episode might actually belong to different classes in a different episode. This requires more flexibility and is a more challenging problem compared to standard few-shot classification. So to me, this experiment of checking for (in)consistency across FSL benchmarks would be more convincing if performed on different benchmarks for the same problem setting, e.g. standard few-shot classification. Still, it should be noted that SVM slightly outperforms RR and PN in one case and slightly underperforms on the other, so this is not a drastic difference either way.

Related work
------------------
- [6] makes a similar separation of the test tasks of Meta-Dataset into ‘weak generalization’ and ‘strong generalization’ sets, where the former introduces held-out classes at test time from *seen* datasets (some other classes of which were used for training), whereas the latter introduces classes of held-out *datasets* altogether (none of their classes used for training). These can also be thought of as a form of ID versus OOD evaluation, and they also show trade-offs between models performing better on one setting but worse on another. It would be useful to mention this connection.

- [7] creates training and test tasks automatically via a clustering framework that allows to control the degree of difficulty (similarity between the resulting sets of training and test classes out of the overall available set of classes). They also show that models’ performance is “flipped” for their “easy” and “hard” settings. Perhaps their framework can be thought of as constructing a spectrum between ID and OOD evaluation.

Minor comments
-----------------------
- In Figure 2c, why is there no curve for ValGen like in the remaining sub-figures of Figure 2?
- RE: Table 1 and lines 206-209, ProtoMAML [2] can be used instead of MAML for handling varying numbers of classes
- Line 126: “mini” → “miniImageNet” ?
- Line 128: “have lead” → “have led”
- Line 392: “set set” → “test set”

References
----------------
- [1] Dynamic few-shot visual learning without forgetting. Gidaris and Komodakis. CVPR 2018.
- [2] Meta-Dataset: A Dataset of Datasets for Learning to Learn from Few Examples. Triantafillou, Zhu, Dumoulin, Lamblin, Evci, Xu, Goroshin, Gelada, Swersky, Manzagol and Larochelle. ICLR 2020.
- [3] A Large-scale Study of Representation Learning with the Visual Task Adaptation Benchmark. Zhai*, Puigcerver*, Kolesnikov*, Ruyssen, Riquelme, Lucic, Djolonga, Pinto, Neumann, Dosovitskiy, Beyer, Bachem, Tschannen, Michalski, Bousquet, Gelly and Houlsby. 2020.
- [4] Comparing Transfer and Meta Learning Approaches on a Unified Few-Shot Classification Benchmark. Dumoulin*, Houlsby*, Evci, Zhai, Goroshin, Gelly and Larochelle. 2021.
- [5] Flexible few-shot learning with contextual similarity. Ren*, Triantafillou*, Wang*, Lucas*, Snell, Pitkow, Tolias, and Zemel. 2020.
- [6] Learning a Universal Template for Few-shot Dataset Generalization. Triantafillou, Larochelle, Zemel and Dumoulin. ICML 2021.
- [7] Embedding Adaptation is Still Needed for Few-Shot Learning. Arnold and Sha. 2021.
- [8] A Broader Study of Cross-Domain Few-Shot Learning. Guo∗, Codella∗, Karlinsky, Codella, Smith, Saenko, Rosing, and Feris. 2020.


**Time Spent Reviewing:**

4

---

> ### Author Response · Authors · 2021-08-10
> **Response to Reviewer Qp31**
>
> **[Performance differences and ID/OOD ranks of meta-learning methods]**
> Please see the common response where we answer this in detail.
>
>
> **[Relationship with low/Incremental few shot]** As we explain in L90-92, low-shot/incremental few shot learning does not have a notion of a training task distribution -- instead they only have a single training task which is to classify all of the base classes simultaneously. Additionally, the performance over this single training task is obtained through supervised training of a single classifier without a notion of meta-learning an algorithm. Thus their setup is different from what we investigate in our work, which focuses on the evaluation of meta-learning algorithms’ generalization alone. Hence we exclude this line of work in our methods comparison.
>
> **[Model selection takeaway]** For ID model selection, we believe using additional held-out examples for seen classes is a natural and canonical way to perform snapshot selection, and this is also what we used for FEMNIST and Zappos-ID datasets. For OOD model selection, we acknowledge that it is still unclear how to best perform snapshot selection because in different scenarios the best strategy might be different. Similar ambiguities have been previously found in domain generalization (L339-342). We hope that our discussion about the existence of such ambiguity in the meta-learning domain as well could motivate the community to collectively reason about and improve the model selection strategy for OOD meta-learning evaluation.
>
> **[Are the conclusion flip results surprising?]** Although it may seem obvious that having a smaller meta-test set would lead to conclusion flips and unreliable evaluations, we analyze both performance flips and exaggerations through a set of rigorous controlled experiments where in addition to size of the novel set of classes we consider other factors like no. of base classes in training, true performance difference on larger set of underlying classes, and size of underlying set of classes which can impact the reliability of conclusions while evaluating meta-learning methods. Despite some of the results confirming our common intuition, we believe our rigorous analyses are still warranted. Additionally, many works in meta-learning theory [1,2,7,20] only experimentally evaluate their claims using the smaller scale-datasets with fewer novel classes. Our findings also highlight the potential unreliability of such practices.
>
> **[Should the narrative be more general and about all possible few-shot learning approaches?]** We focus our narrative on meta-learning evaluation without discussing the transfer learning approaches in detail because the term transfer learning naturally focuses on OOD generalization. In contrast, as we have argued in Section 3 and 4, meta-learning FSL can encompass both ID and OOD generalization. We include transfer learning methods such as supervised-learning baseline (SB) and meta-baseline (MB) only to show that even transfer learning methods cannot necessarily achieve the best of both ID and OOD generalization simultaneously.
>
>
>
> **[Proof-by-contradiction assumption’s realisticness]** Common practice in meta-training uses more than $10^4$ training tasks constructed from the base classes. Here, in your given example, the total number of classes in all the ten datasets in Meta-Dataset is around 5000. Thus our assumption that the number of training tasks $N \ge 10^4 \ge \sqrt{5000} \approx \sqrt{\mathcal{C}_L}$ is still very reasonable. In fact, it is even easier to argue that training on Imagenet and evaluating on all ten datasets in the Meta-Dataset is an OOD scenario -- here, instead of counting the classes, we can think about how many datasets we’ve encountered during training versus during testing. It would be impossible for *iid* task sampling to only see classes from a single dataset during training when at test time we have a significant proportion of classes from many different datasets.
>
>
>
> **[ID vs OOD results]** See common response for significance of ID/OOD conclusion flips. Additionally, the different model selection strategies you suggest for OOD and ID scenarios are exactly the ones we employ.
>
>
>
> **[Ranking similarity analysis's metric]** We believe the Kendall rank correlation coefficient $\rho$ is an appropriate metric to compare rankings in our case since it is very robust to movement of nearby items. For example, given a ranking list of $n$ items, if we only switch the position of two neighboring items, the resulted ranking has a $\rho$ of $1 - \frac{4}{n(n-1)}$ with the original list. Thus this degree of similarity is still very high (close to 1) especially when n is reasonably large (in our case $n=20$ or $40$). Moreover, in all our experiments the set of snapshots we want to select from are exactly the ones at the later stage of training which are all close in their performance values (we would never want to select the snapshot at the very beginning of training as its performances are much worse than the later ones). As a result, all the 20/40 snapshots we considered have a ValGen/NovelGen accuracy difference of roughly $\pm$1%. Thus, we believe your suggested approach of using a "weighted average of snapshot performance" would give us findings similar to what we have already got.
>
>
> **[More realistic ID setting for modified FSL benchmarks]** We emphasize that our ID modification of the current FSL benchmark is already what you think as more compelling. For example, in miniImagenet, because there are 64 base classes, there exists $64 \choose 5$ $\approx$ $7.6 \times 10^6$  different 5-way tasks in this base task distribution. During training, we never sample over 250k tasks from this base task distribution $\mathbb{P}_{\mathcal{C}_B}$. Therefore, when we sample fresh tasks from this base task distribution during ID generalization evaluation -- many of the sampled tasks would already have a class composition that has never been seen during training. We choose not to generate all possible base tasks through enumeration due to the large number; however, what we’ve already done is equivalent to your description of the randomized task partition scheme.
>
>
>
>
>
>
>
> **[Diversity of the novel class set in addition to its size]** It is correct to say that the diversity of the test set also plays a role in truly establishing the algorithm’s OOD generalization ability. However, in our Inconsistency Example 1 analysis setup, we assume the novel classes are uniformly and non-repeatedly sampled from a larger underlying set of classes. This uniform sampling is extremely unlikely to sample tasks with classes only from a very small subcategory (classes being all similar to each other). Therefore, your question cannot directly fit into our analysis framework. However, the Meta-Dataset paper’s own results could already give supporting evidence to your argument. In Table 1 in [38], we see that when trained only on Imagenet, fo-Proto-MAML is a generally better meta-learning method across all datasets combined, but it is still outperformed by fo-MAML on a single specific dataset Textures. If we were to only use Textures to compare these two methods’ OOD generalization performance, we might have arrived at the wrong conclusion.
>
>
>
> **[(In)Consistency across OOD FSL benchmarks]** It is true that mini-OOD and Zappos-OOD reflect different few-shot learning scenarios. However, both of these datasets would still fit into the same category of OOD FSL generalization evaluation. We never claim the performance rankings are inconsistent among the current OOD FSL benchmarks---instead, we claim in L429 - 431 that the performance rankings of different meta-learning methods can possibly flip in different OOD scenarios, depending on the nature of the shift between train and test task distributions. Our observations highlight that comparing meta-learning methods in the entire space of OOD FSL generalization is much more nuanced than ID where the rankings are much more consistent.
>
>
> **[Related work]** We believe the weak generalization and strong generalization in your reference [6] does not map to our definition of ID vs OOD evaluation. Specifically, we have shown that even when the base and novel classes are taken from the same datasets (i.e. weak generalization), as long as they are disjoint with each having a small number of classes compared to the number of sampled tasks, the training and test tasks generated in this scenario still cannot be thought of as coming from the same distribution. Thus both the weak and strong generalization scenarios are still performing OOD evaluation, with the difference being the degree of the task distribution shift being smaller in the weak scenario than the strong scenario. Thus "observing trade-offs between models performing better on one setting but worse on another" is an example of meta-learning methods' ranking inconsistency across different OOD FSL benchmarks (with varying shifts) but not an inconsistency between ID vs OOD scenarios.

---

> > ### Comment · Reviewer_Qp31 · 2021-08-15
> > **response**
> >
> > Thank you for your response and clarifications.
> >
> > **[Relationship with low/Incremental few shot]** I am aware of the differences in the problem setting usually explored on the low-shot benchmarks. But I still think it’s compelling to re-use these benchmarks (e.g. their splits at the very least) instead of creating new ones, since they already have an example split as well as a class split, to examine generalization both to novel classes at test time, as well as to held out examples of base classes at test time. Since this problem is already studied extensively in that community, there are likely some insights there that are relevant to the discussion in this paper.
> >
> > **[Model selection takeaway]** I certainly agree that this ambiguity exists and that it’s useful to point it out. My concern is the potential limited significance / impact of that discussion, in the absence of insights into how to address this problem.
> >
> > **[More realistic ID setting for modified FSL benchmarks]** The difference between this setup and what I was suggesting is that in this setup, some of the tasks seen at test time will contain the same class composition seen at training time. A cleaner setup would ensure that this is never the case, though I understand the argument that *most* of the test tasks will have unseen class combinations.
> >
> > **[(In)Consistency across OOD FSL benchmarks]** I completely agree with the underlying message that comparing models’ performance on the entire space of OOD FSL generalization is hard (or impossible). My point, though, is that the inconsistency between models’ performance on standard few-shot classification versus the flexible setting of [33] is expected. In fact, the problem in [33] is motivated as being one where traditional methods suffer, and reflects a different problem (not just the benchmark or splits). I think it would have been more compelling to demonstrate flips in different OOD FSL benchmarks that are more comparable to each other in terms of the problem formulation. For instance, does simply splitting the classes of known benchmarks differently (i.e. re-assigning which classes belong to the meta-train and meta-test sets) result in (slightly) different performance rankings?
> >
> > **[(In)Consistency across OOD FSL benchmarks]** I understand the viewpoint of the paper that even within a given dataset, holding out a set of classes can lead to an OOD setup, but I don’t think what you quote here is an example of OOD inconsistency. There is definitely a difference in degree of generalization challenge when comparing held-out classes of a seen dataset, to held-out *datasets* altogether, and one can reasonably consider each dataset as forming its own “distribution”. The point of suggesting both of those related works in my original review is to highlight the connections between other papers that have also considered degrees of generalization, perhaps under different definitions of "distribution".
> >
> > In regards to the common response, I liked the experiment that shows that reducing the number of training tasks (which makes the problem harder) amplifies the gap between the methods, making the flip between ID and OOD more pronounced. This is a good sanity check to ensure the results were not just a "fluke" due to the small performance differences. Generally, I agree that many recent SotA methods surpass the previous SotA only slightly, making the discussion in this paper regarding reliability of evaluation indeed timely and emphasizing the need to rethink the evaluation protocol or benchmarks used.
> >
> > Taking a step back, after having read the other reviews too, I continue to see several merits to this paper: the alternative view that FSL benchmarks may actually be OOD, despite being thought of as ID in theoretical analyses, and showcasing performance flips in ID/OOD as defined here. I also find the discussion and thorough experiments of the paper interesting, though sometimes inconclusive, making me question their impact.

---

> > > ### Author Response · Authors · 2021-08-24
> > > **Additional Response to Reviewer Qp31**
> > >
> > > Thank you for your response!
> > >
> > > **[Borrowing the setup of incremental/low-shot learning]** Regarding borrowing the dataset split from incremental few-shot learning, we have used the same class split for miniImagenet as used in [15] and for tieredImagenet as used in [32]. In terms of splitting examples, for more accurate in-distribution performance evaluation we used more examples for each base class than the aforementioned papers. We are not aware of low-shot learning evaluations using datasets CIFAR-FS and FC-100. However, we still provide a modification of these benchmarks for in-distribution evaluations.
> > >
> > > **[Insights from incremental/low-shot learning relevant to our paper]** Regarding comparing the insights from incremental/low-shot learning papers, we believe the ultimate evaluation metric of these papers is the classification performance of the adapted model over the union of all the base classes and a test task’s $n$-way novel classes. However, as we have argued, this is not the evaluation criterion for meta-learning FSL methods. To the best of our knowledge, the incremental/low-shot learning papers haven’t conducted analyses about the possibility of improving novel class prediction performance alone at the cost of base class performance. Can you clarify what type of insights you think may possibly be obtained?
> > >
> > >
> > > **[What are the insights from model selection takeaway?]** We believe our quantitative discussion about the potential unreliability of model selection using validation classes is by itself a valuable insight that many meta-learning researchers are not fully aware of. Beyond pointing out this potential problem, we have also proposed an alternative for snapshot selection by picking the snapshot with the best in-distribution generalization performance. To the best of our knowledge, this strategy is also not commonly considered/used in the meta-learning FSL community. By proposing alternative strategies for model selection, we hope our work will motivate meta-learning methodology researchers to deliberate over not only their methodology design **but also the choice of model selection strategies** to be used during experiment evaluations. Similar suggestions have also been made in domain generalization [16] (see related work L95-98), where the authors argue that it is the responsibility of the designer of the new method to choose which model selection strategy works best for their method. Regarding how to address our identified problem, as we have pointed out in our paper (L332-342) as well as in our previous individual reply **[model selection takeaway]**, there may not exist a one-size-fits-all solution for the model selection strategy problem, and as a result, we think making each researcher aware of this case-by-case subtlety is also an important contribution.
> > >
> > >
> > > **[Are there flips in different OOD FSL benchmarks that are more comparable to each other?]**
> > > In our paper, we showed through Zappos-OOD vs. miniImagnet OOD that the conclusion of which meta-learning method works better in different OOD settings can be different depending on the nature of the task distribution shift (Section 5 L423-428). We believe that the reviewer acknowledges our findings and asks whether there would also be *“flips in different OOD FSL benchmarks that are more comparable to each other in terms of the problem formulation”*? We believe we have compelling evidence that flips can occur in these settings based on the results already provided in our paper, which we explain below.
> > >
> > > Instead of comparing rankings across Zappos-OOD vs miniImagenet OOD, we focus our comparison across the standard OOD FSL benchmarks FC-100, CIFAR-FS and miniImagenet, which are all comparable in terms of problem formulation because of the same training, validation, and test construction mechanism. If we compare the out-of-distribution performances (NovelGen) of meta-learning methods SVM, PN and RR over these datasets, we see that the performance rankings over these methods are still not fully consistent. On CIFAR-FS we see that RR>PN>SVM (Fig 4(a) in Appendix E.1) but on FC-100 we have RR>SVM>PN (Fig 4(b) in Appendix E.1). Additionally, for miniImagenet we observe SVM>RR>PN (Fig 1(a) in main paper Section 4). Note that for each of these comparisons we make sure to evaluate methods using 10,000 test tasks so as to not have overlapping confidence intervals. These discrepancies in performance rankings are clearly indicative of conclusion inconsistencies for meta-learning methods on different OOD FSL benchmarks with similar setups. Additionally, note that FC-100 and CIFAR-FS are even more comparable, since they are simply different re-partitions of the same set of CIFAR-100 classes. Thus, it is even more surprising to see the performance order switch between SVM and PN on these datasets.
> > >
> > >
> > > **[Related work - why do we claim in our individual response that the example you provided is an example of OOD inconsistency?]** We appreciate your suggested related work and will cite these papers and include in our revision our discussion in this rebuttal. However, we are unsure why you don’t think what we quote is an example of OOD inconsistency. We will further clarify our reasoning below and please let us know if this helps with understanding our claim.
> > > We believe you do agree that a disjoint set of base/novel classes taken from the same dataset in the current FSL benchmarks would create a shift between the training and test task distributions. As a result, any performance comparison between different meta-learning methods over this type of **weak generalization** benchmark is a comparison of their OOD generalization ability. On the other hand, when unseen datasets are used to generate tasks during meta-testing, this creates a different (and possibly larger) type of distribution shift and hence any performance comparisons made over this **strong generalization** setup is also a comparison of the meta-learning methods’ OOD generalization ability. As mentioned in your review **Related Work**, there exist tradeoffs between models performing better in one setting (weak generalization) but worse in another (strong generalization). We believe this is evidence that in two different OOD scenarios (weak vs strong) which meta-learning method performs better can be different and thus inconsistent. This finding is similar to the findings we made between the two different OOD scenarios Zappos-OOD vs miniImagenet OOD where the performance rankings are also inconsistent (Section 5 Inconsistency example 2: Consistency across multiple OOD FSL benchmarks).

---

> > > > ### Comment · Reviewer_Qp31 · 2021-08-28
> > > > **response**
> > > >
> > > > Thank you for your response.
> > > >
> > > > [Borrowing the setup of incremental/low-shot learning] Thank you for clarifying that the class splits are taken from [15] and [32], respectively. It might be worth clarifying (or emphasizing) this in the paper, e.g. in the “Modifying OOD FSL benchmarks for ID evaluation” section, as I wasn’t sure if this was the case. Aside from class splits, though, I was referring to using the same methodology as previous incremental/low-shot learning methods to establish *example* splits too for ID evaluation. For example, [15] (in Section 4.2 of their paper) describe a benchmark they use based on ImageNet. They use the images of the *validation set* of the (original) ImageNet dataset for evaluation. The thought was that, since that community has already defined a way of creating both class and example splits, it would be useful to re-use their methodology to facilitate comparisons between models developed for that problem, and the results reported in this paper. I do realize though that the low-shot problem setting is different and the results aren’t directly comparable.
> > > >
> > > > [Related work] I understand the viewpoint of even weak generalization being OOD based on the terminology established in the paper. My point in this particular scenario, though, is that stronger generalization tasks are harder than weak generalization tasks, perhaps we can say that they are “even further out of distribution”, though this remains to be formalized. The other related work I mentioned (reference [7] of my original review) also discussed “degrees” of relatedness between training and test classes; both of these seem like a good addition to the narrative instead of considering only two groups: ID and OOD, since, for instance, there can be varying degrees of difficulty even within the OOD setting. I don’t think we fundamentally disagree here, these are simply some additional data points to add to the discussion as appropriate.
> > > >
> > > > [OOD flips in comparable benchmarks] Agreed that the three benchmarks mentioned in the response are comparable in this sense. However, although these results are interesting, I’m not sure that using only 3 models on a small number of datasets is sufficient to demonstrate this convincingly. The “common response” is helpful in showing that the performance differences are meaningful, especially the additional experiment reported there. But I still think that using a larger number of models and / or a larger number of random partitions of classes (even within a single dataset) would yield more compelling results, in line with comments from other reviewers.
> > > >
> > > > [Overall] Taking a step back, the main contribution of this paper, as I see it, is alerting the community about the difference between ID and OOD evaluation of meta-learning methods, and the inconsistency between discussions and assumptions in theory papers, versus how FSL benchmarks are constructed in practice. I view this as being insightful, useful and interesting. I’m less excited about some of the other analyses in the paper. Specifically, for OOD flips, I think it would be more convincing to use more models/ class splits as I mentioned above. In terms of model selection, I agree it is useful to discuss alternative approaches, but I’m unsure of the degree of significance of the discussion in the absence of a clear take-away message. The unreliability of small test sets is also a good observation, and the experimental analysis is rigorous and interesting, but I’m not sure how impactful this is given that we are already moving towards using larger datasets for FSL evaluation. For these reasons, this paper is borderline in my opinion.

---

> > > > > ### Author Response · Authors · 2021-08-31
> > > > > **Additional Response to Reviewer Qp31**
> > > > >
> > > > > Thank you for your response!
> > > > >
> > > > > **[Borrowing the setup of incremental/low-shot learning]** Thank you for suggesting to emphasize that we use the same class split as [15] and [32]---we will make this more explicit in our revision. In terms of example-level split, for miniImagenet, we use **all the unused ImageNet training examples** from each base class (only 600 images out of a total of approximately 1300 were selected from each base class for miniImagenet meta-training). This is unlike [15] which only uses 300 unused ImageNet training examples from each base class. By using more unseen examples from each base class, we aim to ensure more accurate in-distribution generalization performance estimation. For tieredImagenet, the original authors [32] use every training example from each of the 200 base classes for meta-training. We find the number of examples per base class from the ImageNet validation set to be too small (~50 per base class) to be used for accurate in-distribution generalization evaluation, and so we instead choose to randomly split the examples into an 80%-20% split and use the 80% for meta-training and the remaining 20% for ID evaluation. We use a similar split for CIFAR-FS and FC-100 because to the best of our knowledge there does not exist prior incremental/low-shot learning works which have used these two benchmarks.
> > > > >
> > > > > **[OOD flips in comparable benchmarks]** In our paper, we prove by existence that different OOD benchmarks can have different performance rankings of the same set of meta-learning methods. We believe that the ranking inconsistencies of several popular meta-learning methods over multiple pairs of OOD benchmarks (Zappos-OOD vs mini-OOD), (mini-OOD vs FC-100-OOD), (mini-OOD vs CIFAR-FS-OOD), (FC-100-OOD vs CIFAR-FS-OOD) has already established our claim that the conclusion-inconsistency phenomenon clearly **exists**. Including additional datasets/meta-learning methods would be helpful in further establishing how prevalent this phenomenon is but we don’t believe it is necessary for proof of its existence. The main focus of our paper is to caution meta-learning researchers to be mindful of the scope of a meta-learning method’s improvement as the performance advantage might only be limited to certain specific OOD scenarios but not others, which we believe is sufficiently supported by the existence of our experimental results. Further studying how frequently the conclusion inconsistency exists is an interesting future direction motivated by our findings.

---

### Official Review · Reviewer_qFSe · 2021-07-16

**Rating:** 8
**Confidence:** 3

**Summary:**

This paper analyzes current meta-learning evaluations and finds that there is a significant discrepancy between how they are often theoretically and experimentally studied. They find that most meta-learning benchmarks are constructed with OOD test sets, whereas theory focuses primarily on ID test sets. The authors conduct numerous evaluation to understand the impact of this design oversight.

**Limitations And Societal Impact:**

The authors did not address limitations and societal impact, which I agree with. They are pointing out flaws in the current evaluation methodology.

**Main Review:**

The main strength of this paper is pointing out the difference between OOD and ID test sets for meta-learning, and the following experiments the authors ran on the FSL benchmarks to understand the empirical impact of this difference.

Of particular importance are the ranking experiments the authors run, which finds that methods that work on ID benchmarks can in fact be the worst for OOD benchmarks. This finding is very important for practitioners in this area to be aware of what setting their application is in.

I also liked the experiment where the authors converted existing OOD benchmarks into ID benchmarks and re-ran existing methods on the converted datasets. The high degree of ranking flips is concerning and is another point to show where the theory and experiments diverge for FSL.

The writing exposition was very clear, except that there is a considerable amount of bold and italics throughout the text which hinders reading flow (nit).

I vote to accept its paper based on its clear and practically relevant insight which is much needed for the community, as well as the strong supporting experiments.

**Time Spent Reviewing:**

2

---

> ### Author Response · Authors · 2021-08-10
> **Response to Reviewer qFSe**
>
> We appreciate you recognizing the merit of our paper. We hope to share our unexpected findings about ID and OOD meta-learning evaluation with the entire meta-learning community. Thanks also for your constructive comments regarding the presentation; we will plan to take a pass in our revision to reduce the use of bold/italics so that it is easier to parse.

---

### Official Review · Reviewer_rpPm · 2021-07-17

**Rating:** 6
**Confidence:** 3

**Summary:**

- They explain that in standard meta-learning datasets the evaluation is OOD - the test tasks are not sampled from the same distribution as train tasks. They explain that ID meta-learning is important too.
- They consider 2 ID meta-learning datasets and 1 OOD meta-learning dataset (Table 1). They show the ranks of algorithms on the ID datasets is flipped from the OOD dataset. On the other hand, the 2 ID datasets are consistent.
- They show that the ranks of algorithms are flipped between zappos-OOD and mini-ImageNet-OOD, suggesting evaluation on OOD datasets can have high variance
- They show that the training tasks, validation tasks, and test tasks, can follow different curves during training time, so using each of these for model selection would give a different model
- They explain that meta-learning datasets often have very few validation tasks, which also leads to high variance in the results


**Limitations And Societal Impact:**

Sounds good!

**Main Review:**


Post rebuttal: updated score to 6 (borderline but leaning a bit towards accept), comments below. I'm starting to appreciate the overall message, and it seems useful - I think with better writing (removing less significant pieces, focusing more on the core message) this could be a strong paper.

---------------------------------------------------------------------------------------

Strengths
- It's nice that they examine many different types of inconsistencies: ID vs OOD, difference between OOD datasets, different methods for model selection, inconsistencies due to small number of tasks in the validation set
- I agree with their general conclusion: OOD is a very broad term and can capture all sorts of distribution shifts. It's difficult to get a method that performs well across OOD datasets, so we need to focus on more structured shifts, for example ID meta-learning with a large number of classes

Weaknesses
- As a primarily empirical paper, where the main message is about the inconsistency of OOD meta-learning datasets, it would be good to have more comprehensive evaluations, that is more datasets and methods
- A key argument is that ID meta-learning datasets rank methods more consistently than OOD meta-learning datasets. I'd like to see more datasets to see if this is really the case.
- They say that model ordering can completely flip across different OOD datasets.
One note: the accuracies of these methods are very close. It's natural that when using different datasets you'd get some variation, e.g. by 1%. For example, in table 1, the difference between PN and Ridge is 0.01%. For FO-MAML, where the effect size is larger, the ordering is consistent across datasets.
- Correlation between valgen selection and testgen selection: they showed that the rank correlation coefficient for snapshots on meta-learning datasets is much lower than for supervised learning, and is generally lower than for zappos-id, their ID meta-learning dataset.
- I think some concrete numbers could help me. For each dataset, if you select the best model using val gen for each method, vs select the best model using base gen and test gen, how different are the accuracies on test gen?
- How were hyperparameters tuned for each method? The appendix shows hyperparameter values for each method, but ideally we'd run a grid search and choose the best hyperparameters for each model. Particularly when the accuracy numbers are so close, this could change the results a lot.
- Their actionable insight 1 on page 9, of using more novel classes, is certainly true if there are more novel classes available.
- Actionable insight 2 is interesting - it's not clear to me why training on more base classes decreases the test variance. This could be examined more. Is it because when training on more classes, there is a larger separation between methods, so we need fewer test tasks to rank them? Is it specific to this one dataset (tiered ImageNet), or more general?

I think the paper has useful things to say, and could be compelling with a more thorough investigation of its findings. The paper would be more compelling if they had a couple more ID datasets, and methods, ideally one or two more OOD datasets, and if they grid search over their hyperparameters or run ablations on them.


**Time Spent Reviewing:**

4+2 hours

---

> ### Author Response · Authors · 2021-08-10
> **Response to Reviewer rpPm**
>
> **[More datasets to show ID ranks methods more consistently than OOD]**
>
> - **[Consistency of ID ranks]** Unlike the current FSL benchmarks which as we have argued targets only OOD evaluation, we believe that there does not exist a set of popular standardized datasets that specifically evaluate meta-learning methods’ ID performance. In our work, we take the first steps in this direction and propose realistic datasets that can provide ID evaluations: federated learning FEMNIST handwriting recognition benchmark and the online advertising Zappos-ID attribute classification benchmark. In addition to these datasets (with three different variations in terms of task ways and shots), we have also discussed a way to modify existing OOD FSL benchmarks (mini, CIFAR, FC-100, tiered) to enable ID evaluation. The meta-learning methods’ ID performance ranking is not only consistent over the two realistic scenarios we introduced **but also over each of the four modified datasets**. Although we believe observing ID performance ranking consistency over these six datasets substantiates our claim, we also agree that more standardized ID benchmarks in the future would help verifying it further.
>
> - **[Inconsistency of OOD ranks]** On the other hand, in terms of meta-learning methods' ranking inconsistency over different OOD scenarios, we show i) Ranks on Zappos-OOD are inconsistent with all four common OOD FSL benchmarks; ii) Other works have similarly found that the better meta-learning method (for example TADAM [28] > Protonet) on the common OOD FSL benchmarks can perform worse on a different OOD FSL learning scenario (TADAM < Protonet) (Table 2 in [33]); iii) In Appendix E.3 we show (Figure 5) that on a set of OOD task distributions  $\\{ \mathbb{P}_{\scriptscriptstyle \textrm{BN}}(\lambda), \lambda \in [0,1] \\}$ obtained by interpolating base and novel task distributions of FSL benchmarks, the OOD performance ranks of meta-learning methods can differ a lot based on the degree of shift (given by $\lambda$).
>
>
> **[Performance difference between meta-learning methods]** Please see our common response on performance differences in the ID and OOD scenarios. Regarding your mentioned example, it is indeed true that on FEMNIST, the performance difference between PN and Ridge is not significant. However, we didn’t claim PN to outperform Ridge in this case -- we rank both of them in the first place (superscript (1)).
>
>
>
> **[Concrete numbers of NovelGen difference when using ValGen vs. BaseGen for snapshot selection]** We show for two different datasets in the table below what test performance (NovelGen) we get when the snapshot is chosen using the best BaseGen snapshot (row 1) vs the best ValGen snapshot (row 2), and compare them against the best possible NovelGen across all snapshots (row 3). Here we notice that there can be a significant difference in the selected snapshot’s NovelGen performance if we use one selection strategy instead of another and there isn't a single better snapshot selection strategy.
>
> | NovelGen performance  selection strategy | PN on tiered-M (Fig 2(c)) | SVM on cifar-M (Fig 2(d)) |
> |:------------------------------------------:|:---------------------------:|:---------------------------:|
> | Best BaseGen                             | 80.45%                    | 80.49%                    |
> | Best ValGen                              | 80.05%                    | 82.65%                    |
> | Best NovelGen                            | 80.45%                    | 82.79%                    |
>
>
>
> **[Hyperparameter tuning]** We have run grid search to tune both the meta-learning method-specific hyperparameters and the optimization hyperparameters for each meta-learning method. For OOD FSL benchmarks, we found the best hyperparameters are often the same as what was reported in the original paper. Additionally, the absolute performance and performance orders we report on OOD FSL benchmarks match with other works after we tuned the hyperparameters -- indicating that we have been fair in representing each meta-learning method with its best hyperparameter setting. For our newly identified in-distribution benchmarks, we took care in tuning these hyperparameters for each method to ensure fairness of comparison.
>
>
> **[Actionable insight 2 - understanding why more base classes would decrease conclusion flips]** It is indeed correct that training on more base classes would decrease the variance of the estimator $A(\mathcal{C}_N)$. Here, on the tieredImagenet with 20 random novel classes chosen from the underlying 552, when we train with 64 base classes, the standard deviation of $A(\mathcal{C}_N)$ is 3.1% (Protonet), which is higher than the standard deviation of 2.43% (Protonet) when we train with a superset that has 351 classes. We hypothesize that including more base classes makes the training task distribution more diverse, thus generally decreasing the distribution’s divergence from the randomly sampled novel task distribution (induced by 20 novel classes). As a result, the learned model has better and also more consistent performance over these random novel distributions, resulting in reduced variance. We agree that further understanding this phenomenon is an interesting future direction that our work has motivated.
>
> **[Actionable insight 2 - Can larger separation between meta-learning methods' true performance explain the observation?]** The reviewer hypothesizes that the larger separation between different meta-learning methods’ true performances is the reason for the reduction of conclusion flips when we train with more base classes. However, this is not the case as we intentionally selected the pair of PN, Ridge snapshots that differ in true performance $A(\mathcal{C}_L)$ by the same amount of 0.5% both when we train with 64 base classes and when we train with 351 classes (Fig 3(b) (iv) vs (vi)).

---

> > ### Comment · Reviewer_rpPm · 2021-08-31
> > **Thanks for the response**
> >
> > Thanks for the response, and sorry for the delay. I needed some time to process it and re-read some parts of the paper.
> >
> > [More datasets to show ID ranks methods more consistently than OOD] Ok agreed, there are essentially 6 ID datasets, which sounds good to me. Could you briefly explain why you didn't use FEMNIST OOD, and why Zappos OOD is in a separate table? I'm mostly happy with this, but I'm a little confused about the presentation of the results because they seem scattered everywhere.
> >
> > [Concrete numbers of NovelGen difference when using ValGen vs. BaseGen for snapshot selection] Thanks for the numbers. Can you please update the writing to reflect that ValGen and NovelGen are basically the same (there was something about the ranks switching around a lot, but the effect size seems tiny).
> >
> > [Actionable insight 2 - understanding why more base classes would decrease conclusion flips] Cool thanks, sounds like a good intuition and conjecture.
> >
> > Effect sizes - the common response sounds good. Sorry I missed that part of the paper discussing the effect sizes.
> >
> > Everything else - sounds good.
> >
> > Overall, I think this is a useful paper. I'm a bit borderline but now very slightly leaning towards a borderline accept. The project looks technically solid, but the writing seems confusing - there's a lot going on, and it's hard to extract the key points, which left me confused at many times. This could also just be me, in which case I'm sorry about that!

---

> > > ### Author Response · Authors · 2021-09-02
> > > **Additional Response to Reviewer rpPm**
> > >
> > > No problem, we really appreciate you taking the time to reply and further read the paper.
> > >
> > > **[Why isn’t there a FEMNIST OOD dataset?]** A task $\mathcal{T}_{id}$ in the FEMNIST dataset is specified by a writer $id$. As we discussed in L146-150, any randomized partition of the ~3500 writers/tasks into training and test sets would guarantee that the training and test tasks are drawn *iid* from the same underlying task distribution, thus making the benchmark reflect an ID scenario. In contrast, creating an OOD scenario would require partitioning the writers in a non-random and intentional (possibly adversarial) way. For example, if we had a different handwriting classification dataset where some of the users write in English letters while others write in Japanese characters, we could possibly use this natural partition for training and test tasks to generate an OOD scenario. For FEMNIST, it is not obvious how one would perform such an intentional OOD partition, and the current usage of this dataset in the federated learning literature instead generates training and test splits by randomly sampling users (reflecting the ID scenario).
> > >
> > > **[Why is Zappos OOD in a separate Table (Figure 3(c))?]** We use Zappos OOD in Figure 3(c) to emphasize the inconsistency within the realm of OOD evaluations itself. Previously in Section 4 (Table 1 and Figure 1(a)(b)), we have shown that the performance rankings of the same set of meta-learning methods can possibly be different in the ID vs OOD scenarios. On the other hand, in Section 5.2, we discuss the inconsistency of performance rankings of meta-learning methods over different *types* of OOD benchmarks. Here we compare between two different OOD benchmarks: Zappos OOD vs miniImagenet OOD. Zappos OOD is used here to demonstrate the fact that the performance ranking of meta-learning methods on this OOD dataset (PN > Ridge > SVM) can be flipped from the performance rankings of the same set of methods over another OOD dataset miniImagenet (SVM > Ridge > PN). As Zappos OOD is used to illustrate a different point from that made in Section 4, we only show it in Section 5.2 where it is explicitly referenced.
> > >
> > > **[Is snapshot selection through ValGen really that close to oracle selection through NovelGen?]** We will make our discussion in this response more explicit in the updated writing. For PN on tieredImagenet, the 80.45% - 80.05% = 0.40% is an example of the NovelGen performance loss caused by using the best ValGen for snapshot selection. In other training runs of different meta-learning method/dataset combinations, this performance loss might be even greater. For example, on the FC100 dataset, the NovelGen performance loss is 1.1% for SVM, 1.2% for Ridge, 0.7% for PN and 0.9% for FOMAML if we select the snapshot with the best ValGen performance as opposed to the one with the best NovelGen performance in hindsight. Besides, sometimes a 0.40% NovelGen performance increase might already alter the conclusion of whether a meta-learning method improves over another or not (as the different methods’ performance difference is commonly not large and sometimes < 1%).
> > >
> > > Thanks again and please let us know if we can discuss any additional questions/concerns.

---

> > > > ### Comment · Reviewer_rpPm · 2021-09-02
> > > > **Updated to 6**
> > > >
> > > > I see, all that makes sense. I wasn't familiar with the FEMNIST dataset, and from "By doing this, we can treat the training and
> > > > 174 test tasks as sampled iid from the same task distribution, unlike current FSL benchmarks." and the section title "ID evaluation on FEMNIST." I inferred (incorrectly) that the original dataset was OOD, but being modified for the ID setting. Probably just me, but if you can clarify that in the paper it would be great!
> > > >
> > > > I do think the results are good but the presentation can be improved. Just in care it's useful, we (reviewers) discussed the paper more and these were my final thoughts:
> > > >
> > > > ------------------------------------------------------------------------------------------------------------------------------------------
> > > >
> > > > Their responses sound good (about number of datasets, effect sizes, etc). I'm now pretty borderline overall.
> > > >
> > > > I think the message of the paper is a bit philosophical. If someone truly cared about OOD extrapolation, as they often do in meta-learning, they can't just change the problem and try ID meta-learning. If they're working on ID meta-learning then they should just do that. And some of these OOD inconsistencies have been observed in the past.
> > > >
> > > > So it seems the message is a philosophical one targeted at ML researchers who are developing new algorithms: that the ID setting is important, and may be more tractable because trends are consistent. And more comprehensively showing that results and rankings OOD have a lot of variance.
> > > >
> > > > In general, I'm happy with a message of that sort. I think the presentation of the message and results was below bar (but it could just be me!), which is especially important for more philosophical papers. There's a lot of less important stuff going on that confuses the message (at least in my mind).
> > > >
> > > > So overall - it's a decent paper with a potentially useful message. My opinion of the paper has improved (after the rebuttal) - I think it's borderline and I wouldn't really mind if it gets in or doesn't get in.
> > > >
> > > > Theory: My impression was that theory typically assumes an ID setup because that's much more amenable to analysis. At least when I tried to do a bit of meta-learning theory I was aware that it's not ID, but I didn't have any idea how to analyze an OOD setting.

---

### Author Response · Authors · 2021-08-10
**Common Response**

We thank all the reviewers for your detailed review and comments. We first answer some common questions and then we address the specific comments and questions of each reviewer in a separate response. Please let us know if you have any further questions. Unless separately referenced, the works we cite in the responses can be found in the reference section of our paper.

Here we address the common concern regarding the significance of the performance differences observed in ID and OOD scenarios and thus also the significance of our conclusion that performance orders of meta-learning methods in ID and OOD settings are clearly inconsistent.

**[OOD performance difference’s significance]** We believe 1% (sometimes even 0.5%) improvement can make a significant impact on real-world applications. Newly proposed meta-learning methods often report improvements over previous SOTA methods using the same model architecture by 1.5% or less, e.g, SVM vs PN on CIFAR in Table 2 of [23], R2-D2 vs GNN on miniImagenet in Table 1 of [3]. As these improvements were also measured on OOD performance, we believe our choice to compare SVM > PN by 1.5% and Ridge > PN by 1% (last col of Table 1) is equally valid. Moreover, as it is increasingly difficult to have new methods that significantly outperform existing methods by a large margin (more than 1.5%), we believe our analyses to check for the reliability and consistency of the performance rankings over methods like PN, Ridge, and SVM are timely and necessary.

**[ID performance difference’s significance]** On the other hand, it is worth noting that the actual ID performance difference among different meta-learning methods depends on the properties of the task distribution. While there exist scenarios where the performance difference is <1%, we would like to remind that there still exist multiple scenarios where the ID performance order is not only flipped (from OOD performance order), but also with a difference >1%: e.g., i) on Zappos-ID 2w5s, PN outperforms SVM by more than 1.4%.  This difference is significant compared to the 95% confidence interval $\approx \pm 0.15$%; ii) On all of the 4 modified FSL benchmarks (Figure 1 (a)(b), Appendix Figure 4 (a)(b)), the in-distribution performance (BaseGen) gap is even wider, with PN outperforming SVM by more than 2% despite being outperformed by SVM by at least 1% on the OOD side.

**[Fewer training tasks can enlarge ID performance difference]** A particular parameter that can affect in-distribution performance differences is the number of tasks used for training. For example, in Zappos-ID 2w5s if we lower the number of training tasks from 1000 to 50, these differences are amplified even further (see table below). Here we see that PN outperforms SVM by greater than 3.5%, strongly demonstrating the unexpected performance flip from OOD to ID scenario. We will clarify this in Section 4.

| Methods | Zappos-ID 2w5s with 50 Train Tasks | Zappos-ID 2w5s with 1000 Train Tasks |
|:---------:|:------------------------------------:|:--------------------------------------:|
| PN      | $^{\mathbf{(1)}} 77.67 \pm 0.17$%         | $^{\mathbf{(1)}} 86.58 \pm 0.15$%           |
| Ridge   | $^{\mathbf{(2)}}74.75 \pm 0.16$%          | $^{\mathbf{(2)}}85.56  \pm  0.16$%          |
| SVM     | $^{\mathbf{(3)}}74.06 \pm 0.17$%          | $^{\mathbf{(3)}}85.12 \pm 0.16$%            |
| FO-MAML | $^{\mathbf{(4)}}69.85 \pm 0.18$%            | $^{\mathbf{(4)}}80.14 \pm 0.15$%            |

---

### Decision · Program_Chairs · 2021-09-28

**Decision:**

Accept (Poster)

**Comment:**

The submission discusses the properties of in-distribution (ID, i.e. same task distribution) and out-of-distribution (OOD, i.e. subject to a task distribution shift) meta-learning evaluation. It contends that current few-shot classification benchmarks reflect OOD performance, and that approaches which perform well in the OOD setting may not necessarily perform well in the ID setting. Finally, it highlights model selection pitfalls and issues with the consistency of meta-learning performance comparisons.

Reviewers found that the submission raises interesting questions, as evidenced by their engagement in the discussions. They agree that the discussion surrounding ID and OOD evaluation for few-shot classification is insightful and of value to the research community, especially since a lot of existing theoretical work makes an ID assumption. Other claimed contributions are less appealing to some reviewers:
- The model selection results appear inconclusive.
- The research community is already moving towards larger and more diverse benchmarks, which lessens the impact of the observation that small test sets are unreliable.
- The 6 common ID datasets and 4 common OOD datasets and the number of approaches investigated remains a small sample size to support the claims made in the paper according to some reviewers.

Reviewers weigh these strengths and weaknesses differently, and resultingly their opinion is split on acceptance. I read the paper to form an independent opinion and think that it raises interesting questions, even if imperfectly. In any case, I don't believe any reviewer is _indifferent_ to it, which to me is a good sign that its publication would have an impact on the few-shot learning community. I therefore recommend acceptance.

**Consistency Experiment:**

NeurIPS has a long history of experimentation. In 2014, NeurIPS ran an experiment in which 10% of submissions were reviewed by two independent committees to quantify the randomness in the review process. This year, we repeated a variant of this experiment to see how the quality of the review process has changed over time.  This paper was part of the experiment and was therefore assigned to two committees (consisting of reviewers, an Area Chair, and a Senior Area Chair) that reached independent decisions.  If both committees made the same recommendation, this recommendation was followed. If a single committee recommended acceptance, the paper was accepted (with the exception of a few cases in which the other committee identified what we considered a fatal flaw, e.g., an error in a key result).

Both committees reached the same decision: **Accept (Poster)**

The other committee assigned to the paper recommended **Accept (Poster)**.  You can find the other set of reviews, along with any follow up discussion with the authors here:
https://openreview.net/forum?id=X4JfcKvvDOw